# Aubergine and piRNAs promote germline stem cell self-renewal by repressing the proto-oncogene *Cbl*

Patricia Rojas-Ríos, Aymeric Chartier, Stéphanie Pierson & Martine Simonelig[*]

## Abstract

PIWI proteins play essential roles in germ cells and stem cell lineages. In *Drosophila*, Piwi is required in somatic niche cells and germline stem cells (GSCs) to support GSC self-renewal and differentiation. Whether and how other PIWI proteins are involved in GSC biology remains unknown. Here, we show that Aubergine (Aub), another PIWI protein, is intrinsically required in GSCs for their self-renewal and differentiation. Aub needs to be loaded with piRNAs to control GSC self-renewal and acts through direct mRNA regulation. We identify the *Cbl* proto-oncogene, a regulator of mammalian hematopoietic stem cells, as a novel GSC differentiation factor. Aub stimulates GSC self-renewal by repressing *Cbl* mRNA translation and does so in part through recruitment of the CCR4-NOT complex. This study reveals the role of piRNAs and PIWI proteins in controlling stem cell homeostasis via translational repression and highlights piRNAs as major post-transcriptional regulators in key developmental decisions.

**Keywords** CCR4-NOT deadenylation complex; germline stem cells; piRNAs; PIWI proteins; translational control
**Subject Categories** Development & Differentiation; RNA Biology; Stem Cells
**The EMBO Journal (2017) 36: 3194–3211**

## Introduction

The regulation of gene expression at the mRNA level is fundamental for many biological and developmental processes. In recent years, Piwi-interacting RNAs (piRNAs) have emerged as novel key players in the regulation of gene expression at the mRNA level in several models. These 23- to 30-nucleotide-long non-coding RNAs are loaded into specific Argonaute proteins, the PIWI proteins (Ishizu *et al*, 2012; Guzzardo *et al*, 2013). Classically, piRNAs repress transposable element expression and transposition in the germline. They are largely produced from transposable element sequences and target transposable element mRNAs by complementarity, which induces their cleavage through the endonuclease activity of PIWI proteins and represses their expression.

Recent studies have shown that piRNAs also target protein-coding mRNAs, leading to their repression by PIWI-dependent mRNA cleavage or via the recruitment of the CCR4-NOT deadenylation complex. This regulation is required for embryonic patterning in *Drosophila* (Rouget *et al*, 2010; Barckmann *et al*, 2015), sex determination in *Bombyx mori* (Kiuchi *et al*, 2014), and degradation of spermiogenic mRNAs in mouse sperm (Gou *et al*, 2014; Goh *et al*, 2015; Watanabe *et al*, 2015; Zhang *et al*, 2015).

piRNAs involved in the regulation of protein-coding mRNAs in *Drosophila* embryos are produced in the female germline and provided maternally. An open question is whether this function of the piRNA pathway in post-transcriptional control of gene expression plays a role in the biology of germ cells and germline stem cells (GSCs). In the *Drosophila* ovary, two to three GSCs are localized in the anterior-most region of each ovariole and self-renew throughout the adult life, giving rise to all germ cells. GSCs in contact with somatic niche cells divide asymmetrically to produce a new stem cell that remains in contact with niche cells (self-renewal) and another cell that differentiates into a cystoblast, upon losing the contact with the niche. Subsequently, the cystoblast undergoes four rounds of synchronous division with incomplete cytokinesis to produce a cyst of 16 interconnected germ cells, of which one cell is specified as the oocyte and the other 15 cells become nurse cells (Fig 1A).

Two features make Piwi unique with respect to the other two *Drosophila* PIWI proteins, Aubergine (Aub) and Argonaute 3 (Ago3). First, it represses transposable elements at the transcription level through a nuclear function, whereas Aub and Ago3 act by endonucleolytic cleavage of transposable element mRNAs in the cytoplasm; and second, it plays a role in the somatic and germ cells of the ovary, whereas *aub* and *ago3* function is restricted to germ cells. *piwi* function in GSC biology has long been addressed. *piwi* is required in somatic escort cells (which surround GSCs) for GSC differentiation, as well as intrinsically in GSCs for their maintenance and differentiation (Cox *et al*, 1998, 2000; Jin *et al*, 2013; Ma *et al*, 2014). One molecular mechanism underlying Piwi function in GSC biology has recently been proposed to involve its direct interaction with Polycomb-group proteins of the PRC2 complex, leading to indirect massive gene deregulation through reduced PRC2 binding to chromatin (Peng *et al*, 2016). Regulation of *c-Fos* by Piwi at the mRNA level in somatic niche cells has also been reported to contribute to the role of Piwi in GSC maintenance and differentiation (Klein *et al*, 2016).

mRNA Regulation and Development, Institute of Human Genetics, UMR9002 CNRS-Université de Montpellier, Montpellier Cedex 5, France
*Corresponding author. Tel: +33 4 34 35 99 59; E-mail: martine.simonelig@igh.cnrs.fr

Translational control acting intrinsically in GSCs plays a major role in the switch between self-renewal and differentiation. Two molecular pathways ensure GSC self-renewal through translational repression of differentiation factor mRNAs: the microRNA pathway, and the translational repressors Nanos (Nos) and Pumilio (Pum; Slaidina & Lehmann, 2014). Nos and Pum bind to and repress the translation of mRNAs that encode the differentiation factors Brain tumor (Brat) and Mei-P26, through the recruitment of the CCR4-NOT deadenylation complex (Harris *et al*, 2011; Joly *et al*, 2013). In turn, cystoblast differentiation depends on Bag of marbles (Bam), the major differentiation factor forming a complex with Mei-P26, Sex lethal (Sxl), and Benign gonial cell neoplasm (Bgcn) to repress *nos* mRNA translation; Pum interacts with Brat in these cells to repress the translation of mRNAs encoding self-renewal factors (Li *et al*, 2009b, 2012, 2013; Harris *et al*, 2011; Chau *et al*, 2012).

Aub has a distinctive role in protein-coding mRNA regulation. In the early embryo, Aub binds several hundred maternal mRNAs in a piRNA-dependent manner and induces the decay of a large number of them during the maternal-to-zygotic transition (Barckmann *et al*, 2015). Aub-dependent unstable mRNAs are degraded in the somatic part of the embryo and stabilized in the germ plasm. These mRNAs encode germ cell determinants, indicating an important function of Aub in embryonic patterning and germ cell development. Indeed, Aub recruits the CCR4 deadenylase to *nos* mRNA and contributes to its deadenylation and translational repression in the somatic part of the embryo. This Aub-dependent repression of *nos* mRNA is involved in embryonic patterning (Rouget *et al*, 2010).

Here, we address the role of *aub* in GSC biology. We show that *aub* is autonomously required in GSCs for their self-renewal. This *aub* function is independent of *bam* transcriptional repression in the GSCs and partly independent of activation of the Chk2-dependent DNA damage checkpoint. Aub is also involved in GSC differentiation; *aub* mutant defect in GSC differentiation is less frequent and involves the Chk2-dependent DNA damage checkpoint. Using an Aub point-mutant form that cannot load piRNAs, we show that piRNAs are required for GSC self-renewal. Genetic and physical interactions indicate that Aub function in GSCs involves interaction with the CCR4-NOT deadenylation complex. Importantly, we identify *Casitas B-cell lymphoma* (*Cbl*) mRNA as a target of Aub in GSCs. *Cbl* acts either as a tumor suppressor or a proto-oncogene depending on its mutations, which lead to myeloid malignancies in humans (Sanada *et al*, 2009). *Cbl* encodes an E3 ubiquitin ligase that negatively regulates signal transduction of tyrosine kinases; it plays a role in hematopoietic stem cell homeostasis, maintaining quiescence, and preventing exhaustion of the stem cell pool (An *et al*, 2015). We show that Aub acts to maintain a low level of Cbl protein in GSCs and that this repression of *Cbl* mRNA by Aub is essential for GSC self-renewal. Furthermore, we find that *Cbl* is required for GSC differentiation, thereby identifying a role for Cbl in the regulation of yet another stem cell lineage.

This study reveals the function of Aub and piRNAs in GSC self-renewal through the translational repression of *Cbl* mRNA, thus highlighting the role of the piRNA pathway as a major post-transcriptional regulator of gene expression in key developmental decisions.

## Results

### *Aub* is intrinsically required in GSCs for their self-renewal and differentiation

Aub and Ago3 are expressed in GSCs, and we addressed their function in GSC biology (Brennecke *et al*, 2007; Gunawardane *et al*, 2007). GSCs can be recognized by their anterior localization in the germarium as well as the presence of the spectrosome, an anteriorly localized spherical organelle in contact with the niche, which is enriched in cytoskeletal proteins (Fig 1A). Cystoblasts also contain a spectrosome that is randomly located in the cell, whereas cells in differentiating cysts are connected by the fusome, a branched structure derived from the spectrosome (Fig 1A). GSCs and differentiating cells were analyzed by immunostaining with an anti-Hts antibody that labels the spectrosome and fusome, and anti-Vasa, a marker of germ cells.

We used $aub^{HN2}$ and $aub^{QC42}$ strong or null alleles (Schupbach & Wieschaus, 1991) to address the role of Aub in GSC biology. Immunostaining of ovaries with anti-Hts and anti-Vasa revealed strong defects in both GSC self-renewal and differentiation, in $aub^{HN2/QC42}$ mutant ovaries at 7, 14, and 21 days. A large proportion of $aub^{HN2/QC42}$ germaria had 0–1 GSC indicating GSC loss, and this defect increased over time (Fig 1B, C and F). A lower proportion of germaria showed differentiation defects, observed as tumors containing undifferentiated cells with spectrosomes (Fig 1D). This phenotype did not markedly increase with time (Fig 1F). Both $aub^{HN2/QC42}$ phenotypes were almost completely rescued following expression of GFP-Aub with the germline driver *nos-Gal4*, indicating that both defects were due to *aub* loss of function in germ cells (Fig 1E and F).

Because GSC loss was the most prominent defect in *aub* mutant ovaries, we focused on this phenotype. We used clonal analysis as an independent evidence to confirm the intrinsic role of *aub* in GSCs for their self-renewal. Wild-type and *aub* mutant clonal GSCs were generated using the FLP-mediated FRT recombination system (Golic & Lindquist, 1989) and quantified at three time points after clone induction. We first verified that *aub* clonal GSCs did not express Aub (Fig EV1A and A'). The percentage of germaria with wild-type clonal GSCs was stable over time (Fig 1G and I). In contrast, the percentage of germaria with *aub* mutant clonal GSCs strongly decreased with increasing time after clone induction, showing that *aub* mutant GSCs cannot self-renew (Fig 1H and I). The presence of *aub* mutant clonal differentiated cysts marked with fusomes indicated that *aub* mutant GSCs were lost by differentiation (Fig 1H). To confirm this conclusion, we used anti-cleaved Caspase 3 staining to record cell death and address whether the loss of *aub* mutant GSCs could be due to apoptosis. The number of GSCs expressing cleaved Caspase 3 was low and similar in control ($aub^{HN2/+}$) and *aub* mutant GSCs (Fig EV1B–D), indicating that *aub* mutant GSCs did not undergo cell death.

Next, we asked whether the GSC self-renewal defect in *aub* mutant ovaries could result from Bam expression in GSCs. Anti-Bam immunostaining of *aub* mutant ovaries demonstrated that *aub* mutant GSCs did not express Bam (100% *n* = 90, Fig EV1E–F').

Finally, we determined the division rate of $aub^{HN2}$ GSCs by counting the number of cysts produced by a clonal marked mutant GSC and dividing it by the number of cysts produced by a control unmarked GSC in the same germarium (Jin & Xie, 2007). As expected, the division rate of wild-type GSCs (FRT40A

**Figure 1.   Intrinsic role of Aub in GSC self-renewal and differentiation.**

A      Schematic diagram of a germarium showing the somatic cells (blue) and the germline cells (green). The spectrosomes and fusomes are shown in orange. The different regions of the germarium are indicated. Region 1: dividing cysts; region 2: selection of the oocyte; region 3: egg chamber with posteriorly localized oocyte. GSCs, germline stem cells; CB, cystoblast.

B–E    Immunostaining of germaria from 7-day-old females with anti-Vasa (green, B–D) or anti-GFP (green, E), and anti-Hts (red). DAPI (blue) was used to visualize DNA. (B) $aub^{HN2/+}$ was used as a control. (C, D) Examples of $aub^{HN2/QC42}$ germ cell loss and tumor, respectively. (E) Phenotypic rescue of $aub^{HN2/QC42}$ with *UASp-GFP-Aub* expressed using *nos-Gal4*. White arrowheads indicate GSCs; the white arrow indicates GSC loss.

F      Quantification of mutant germaria with 0–1 GSC, or with GSC tumors shown in (B–E). The number of scored germaria (*n*) is indicated on the right graph. Error bars represent standard deviation. ***$P$-value < 0.001, ns, non-significant, using the $\chi^2$ test.

G–H′   Germaria containing control (G, G′) or $aub^{HN2}$ mutant (H, H′) clonal GSCs stained with anti-GFP (green) and anti-Hts (red), 14 days after clone induction. DAPI (blue) was used to stain DNA. Clonal cells are marked by the lack of GFP. Clonal GSCs and cysts are outlined with dashed line. White arrowheads show clonal GSCs in the control. *aub* mutant clonal GSCs have been lost (H, H′).

I      Quantification of germaria containing at least one clonal GSC at 7, 14, and 21 days after clonal induction. 50 to 219 germaria were analyzed per condition. Error bars represent standard deviation.

J      Division rate of wild-type and $aub^{HN2}$ clonal GSCs. The number of scored germaria (*n*) is indicated. Error bars represent standard deviation. ***$P$-value < 0.001 using the two-tailed Student's *t*-test.

Data information: Scale bars: 10 μm in (B–E) and (G–H′).

chromosome) was close to 1 (0.95), whereas that of $aub^{HN2}$ mutant GSCs was 0.67, indicating a slower division rate in *aub* mutant GSCs (Fig 1J).

To address the role of Ago3 in GSC biology, we used *ago3* mutant alleles that contain premature stop codons, $ago3^{t1}$, $ago3^{t2}$, and $ago3^{t3}$ (Li *et al*, 2009a). No GSC loss was recorded in the mutant combination $ago3^{t2}/t3$ in 7-, 14-, or 21-day-old females, showing that GSC self-renewal was not affected in the *ago3* mutant. We checked that Aub expression was similar in wild-type and *ago3* mutant GSCs (Fig EV2A–B′). In contrast, $ago3^{t2/t3}$ females showed a prominent defect in GSC differentiation, with a large proportion of germaria having a higher number of undifferentiated germ cells with a spectrosome (two to four in control germaria, versus six to nine in $ago3^{t2/t3}$ germaria; Fig EV2C–F).

Together, these results demonstrate that Aub is required intrinsically in GSCs for their self-renewal and differentiation. Aub maintains GSCs by preventing their differentiation independently of Bam expression. In contrast, Ago3 is specifically involved in GSC differentiation.

### Aub function in GSC self-renewal is partly independent of Chk2

The Chk2-dependent DNA damage checkpoint is activated in several piRNA pathway mutants, leading to developmental defects during mid-oogenesis and, in turn, defective dorsoventral and anteroposterior embryonic patterning (Klattenhoff *et al*, 2007; Pane *et al*, 2007). These developmental defects are partially rescued in double mutants of Chk2 kinase (*mnk* mutant) and different piRNA pathway components. DNA damage in piRNA pathway mutants is thought to result from transposable element transposition. Mobilization of *P*-elements in crosses that induce hybrid dysgenesis (*i.e.,* the crossing of females devoid of *P*-elements with males that contain *P*-elements) leads to a block in GSC differentiation, which is partially rescued by mutation in Chk2 (Rangan *et al*, 2011). To address whether the defects in GSC self-renewal and differentiation in *aub* mutant ovaries might be due to activation of the Chk2-dependent checkpoint, we analyzed $mnk^{p6}$ $aub^{HN2/QC42}$ double-mutant ovaries using anti-Hts and anti-Vasa immunostaining. The *aub* mutant tumor phenotype of undifferentiated cell accumulation was almost completely rescued by $mnk^{p6}$, demonstrating that this phenotype depended on Chk2 (Fig 2A–C). In contrast, the GSC loss phenotype was only partially rescued by

$mnk^{p6}$ and remained visible in a large proportion of double mutant germaria (Figs 2A–C and EV1G), showing that this defect was in part independent of Chk2 checkpoint activation.

These results reveal that the mild defect in GSC differentiation in the *aub* mutant is due to activation of the DNA damage checkpoint. In contrast, the prominent GSC self-renewal defect is partly independent of the DNA damage checkpoint activation, consistent with an additional more direct role of Aub in GSC self-renewal.

### Aub loading with piRNAs is required for GSC self-renewal

To determine whether the role of Aub in GSC self-renewal depends on its loading with piRNAs, we used an Aub double point mutant in the PAZ domain that is unable to bind piRNAs (Aub^AA; Barckmann *et al*, 2015). In contrast to *UASp-GFP-Aub,* which was able to rescue the GSC loss phenotype in *aub* mutant flies when expressed with *nos-Gal4* (Fig 1E and F), expression of *UASp-GFP-Aub^AA* at similar levels (Fig EV3A–C) failed to rescue this phenotype (Fig 2D–G). These data indicate that Aub loading with piRNAs is required for GSC self-renewal. To confirm this result, we used a piRNA pathway mutant in which piRNA biogenesis is strongly compromised. In the absence of Ago3, piRNAs are produced through Aub/Aub homotypic ping-pong (Li *et al*, 2009a) and these piRNAs could be used in Aub-dependent regulation in GSCs, consistent with the lack of GSC self-renewal defect in *ago3* mutant. In contrast, *armitage* (*armi*) encodes an RNA helicase that is essential for piRNA biogenesis (Cook *et al*, 2004; Malone *et al*, 2009). Immunostaining of $armi^{1/72}$ ovaries with anti-Hts and anti-Vasa revealed a GSC loss that increased with time (Fig EV3D–F), showing a function for *armi* in GSC self-renewal.

We conclude that Aub function in GSC self-renewal depends on its loading with piRNAs.

### Aub interacts with the CCR4-NOT complex for GSC self-renewal

Previous reports have shown that PIWI proteins can recruit the CCR4-NOT deadenylation complex to repress mRNAs at the post-transcriptional level (Rouget *et al*, 2010; Gou *et al*, 2014). In the *Drosophila* embryo, Aub is in complex with CCR4, independently of RNA (Rouget *et al*, 2010). To address whether Aub might act through a similar mode of action in GSCs, we analyzed Aub and CCR4 colocalization. CCR4 is present diffusely in the cytoplasm

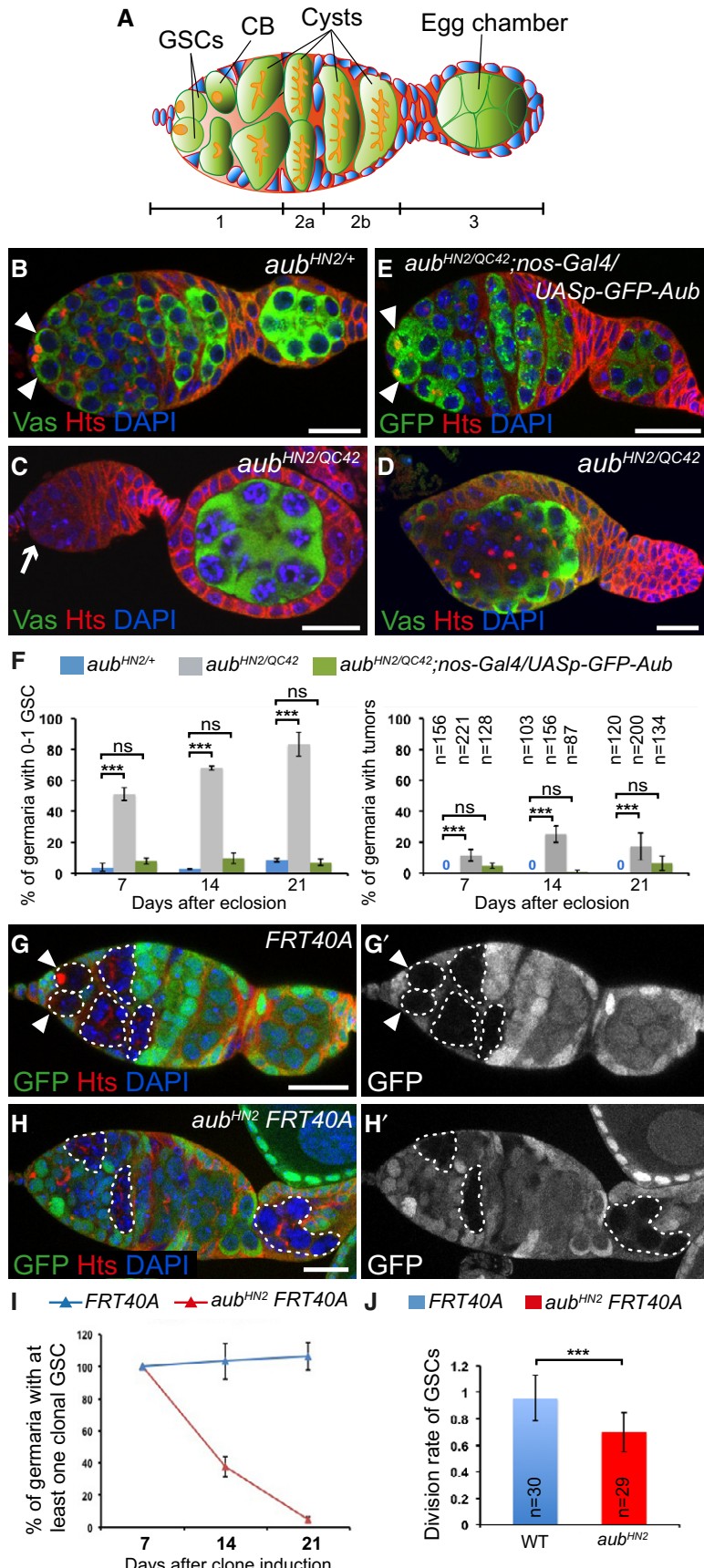

Figure 1.

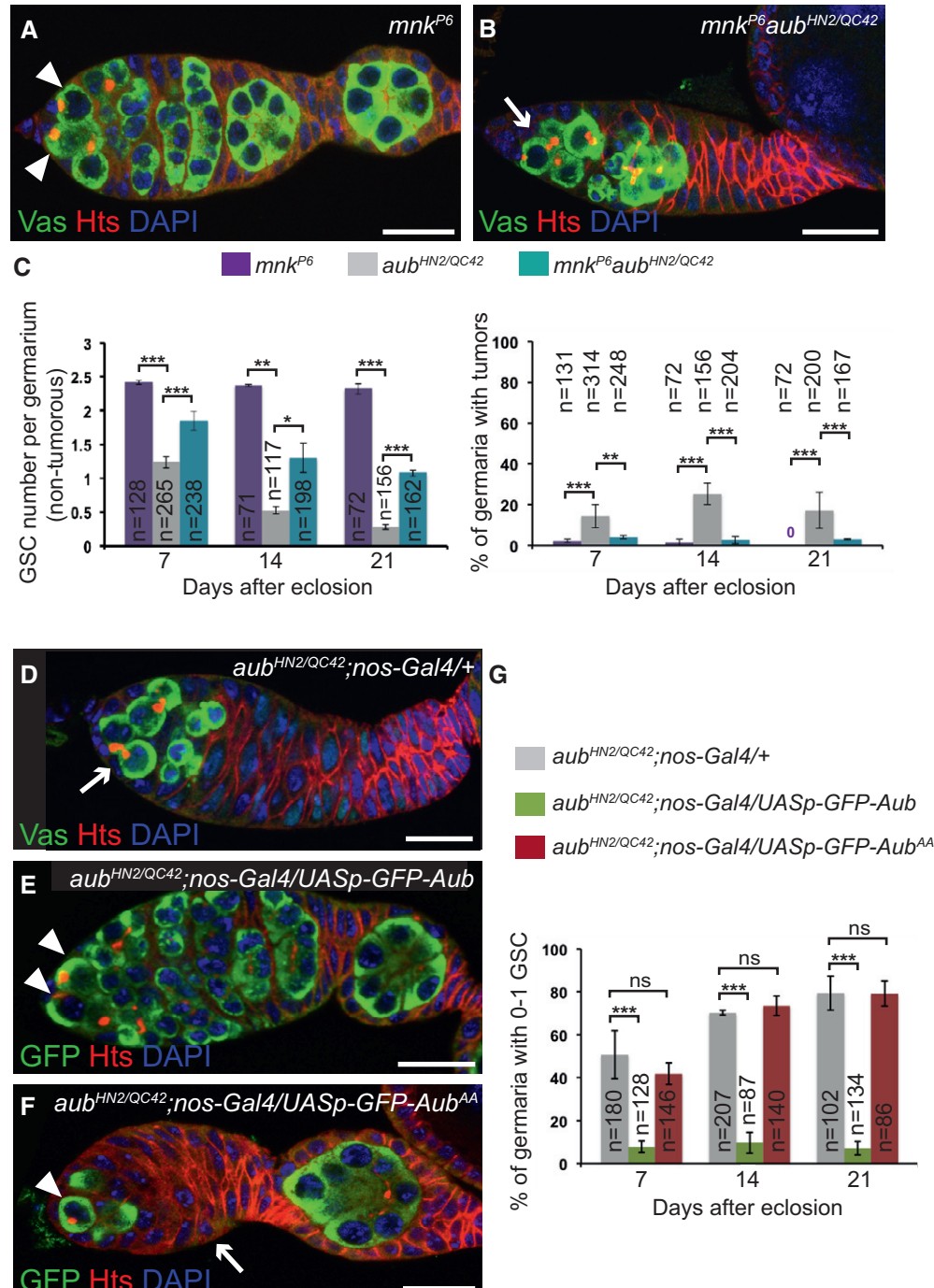

**Figure 2.  The role of Aub in GSC self-renewal is partially independent of Chk2 and requires its loading with piRNAs.**

A, B    Immunostaining of germaria with anti-Vasa (green) and anti-Hts (red). DAPI (blue) was used to visualize DNA. Examples of *mnk*[P6] and *mnk*[P6] *aub*[HN2/QC42] germaria are shown. White arrowheads indicate GSCs; the white arrow indicates GSC loss.

C    Quantification of the number of GSCs per non-tumorous germarium, and of germaria with GSC tumors, in the indicated genotypes. The number of scored germaria (*n*) is indicated. Error bars represent standard deviation. The two-tailed Student's *t*-test and the $\chi^2$ test were used in the left and right panels, respectively. ***P-value < 0.001, **P-value < 0.01, *P-value < 0.05.

D–F    Immunostaining of germaria from 7-day-old females with anti-Vasa or anti-GFP (green) and anti-Hts (red). DAPI (blue) was used to visualize DNA. *aub*[HN2/QC42]; *nos-Gal4/+* was used as a negative control. Examples of rescue in *aub*[HN2/QC42]; *nos-Gal4/UASp-GFP-Aub* germarium (E), and of lack of rescue in *aub*[HN2/QC42]; *nos-Gal4/UASp-GFP-Aub*[AA] germarium (F). White arrowheads indicate GSCs; white arrows indicate GSC loss in (D) and germ cell loss in (F).

G    Quantification of mutant germaria with 0–1 GSC shown in (D–F). The number of scored germaria (*n*) is indicated. Error bars represent standard deviation. ***P-value < 0.001, ns, non-significant, using the $\chi^2$ test.

Data information: Scale bar: 10 μm in (A, B) and (D–F).

    

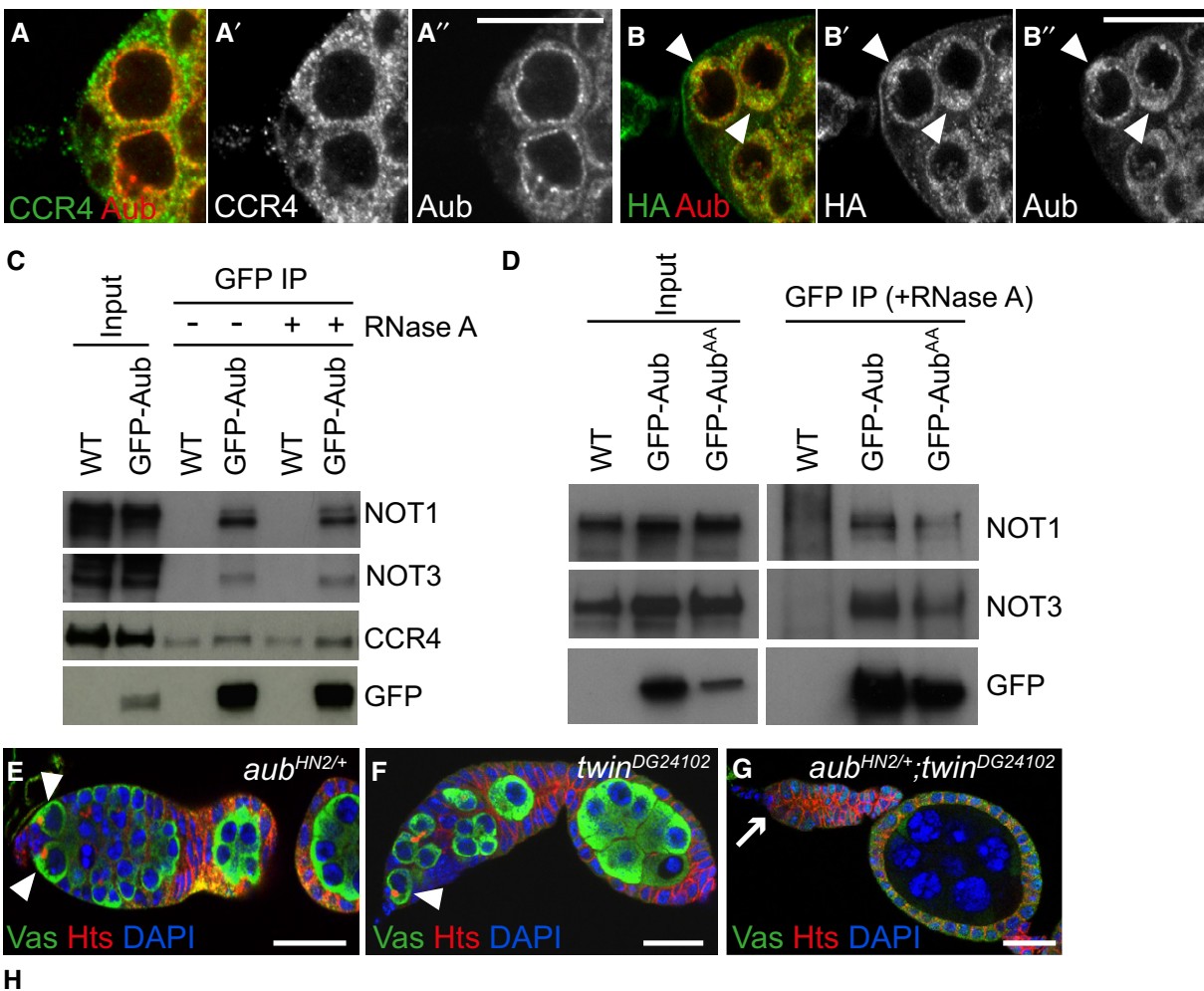

**Figure 3.  Physical and genetic interaction between Aub and the CCR4-NOT complex.**

A–B"   Immunostaining of wild-type (A–A") or *nos-Gal/UASp-CCR4-HA* (B–B") germaria with anti-Aub (red), and anti-CCR4 or anti-HA (green). GSCs are shown in (A–A").
        White arrowheads in (B–B") indicate cytoplasmic accumulation of CCR4-HA colocalized with Aub in GSCs.

C       Co-immunoprecipitation (IP) of NOT1, NOT3, and CCR4 with GFP-Aub in ovaries. Wild-type (WT, mock IP) or *nos-Gal4/UASp-GFP-Aub* (GFP-Aub) ovarian extracts
        were immunoprecipitated with anti-GFP, either in the absence or the presence of RNase A. Western blots were revealed with anti-GFP, anti-NOT1, anti-NOT3, and
        anti-CCR4. Inputs are extracts prior to IP.

D       Co-IP of NOT1 and NOT3 with GFP-Aub or GFP-Aub^AA in ovaries. Wild-type (WT, mock IP), *nos-Gal4/UASp-GFP-Aub* (GFP-Aub), or *nos-Gal4/UASp-GFP-Aub^AA* (GFP-
        Aub^AA) ovarian extracts were immunoprecipitated with anti-GFP in the presence of RNase A. Western blots were revealed with anti-GFP, anti-NOT1, and anti-
        NOT3. Inputs are extracts prior to IP.

E–G     Genetic interaction between *aub* and *twin* in GSC self-renewal. Immunostaining of germaria with anti-Vasa (green) and anti-Hts (red). DAPI (blue) was used to visualize
        DNA. Examples of *aub^HN2/+*, *twin^DG24102* and *aub^HN2/+; twin^DG24102* germaria are shown. White arrowheads indicate GSCs; the white arrow indicates GSC loss.

H       Quantification of mutant germaria with no GSC in 3-, 7-, and 14-day-old females of the genotypes shown in (D–F). The number of scored germaria (*n*) is indicated.
        ***P-value < 0.001 using the $\chi^2$ test.

Data information: Scale bars: 10 μm in (A–B") and (E–G).
Source data are available online for this figure.

and accumulates in cytoplasmic foci, in GSCs (Joly *et al*, 2013). Aub also has a diffuse distribution in the cytoplasm and is present in foci that surround the nucleus collectively referred to as "nuage" (Harris & Macdonald, 2001). Colocalization occurred in diffusely distributed pools of proteins and occasionally in foci (Fig 3A–A''), consistent with deadenylation not taking place in foci (Joly *et al*, 2013). We then overexpressed CCR4-HA in GSCs using *nos-Gal4* and found that CCR4-HA was able to recruit Aub in discrete cytoplasmic regions where CCR4-HA had accumulated, consistent with the presence of CCR4 and Aub in the same complex in GSCs (Fig 3B–B''). Coimmunoprecipitation experiments in ovaries revealed that GFP-Aub was able to coprecipitate the NOT1, NOT3, and CCR4 subunits of the CCR4-NOT deadenylation complex, either in the presence or absence of RNA (Fig 3C). We checked that the coprecipitation of CCR4-NOT subunits was maintained with the mutant form of Aub that does not load piRNAs, GFP-Aub[AA] (Fig 3D).

The *twin* gene that encodes the CCR4 deadenylase is essential for GSC self-renewal (Joly *et al*, 2013; Fu *et al*, 2015). To genetically determine whether *aub* acts together with *twin* in GSC self-renewal, we tested whether GSC loss in the hypomorphic allele *twin*[DG24102] (Joly *et al*, 2013) might be enhanced by reducing the gene dosage of *aub*. GSC loss in *twin*[DG24102] was accelerated in the presence of heterozygous *aub*[HN2] or *aub*[QC42] mutations, consistent with a role for Aub and CCR4 in the same molecular pathway for GSC self-renewal (Fig 3E–H).

Together, these results show that Aub and the CCR4-NOT complex physically interact in GSCs, and cooperate for GSC self-renewal.

### Cbl is an mRNA target of Aub in GSCs

To identify mRNA targets of Aub in GSCs, we looked for candidate genes with a reported role in GSC biology or other stem cell lineages (Appendix Fig S1A). Eight genes were selected, five of which produce mRNAs that directly interact with Aub in embryos (Barckmann *et al*, 2015). Antibody staining in ovaries containing clonal *aub* mutant GSCs was used to record potential increased levels of the corresponding proteins in mutant GSCs as compared to control (Appendix Fig S1A–C''). We found a mild increase in Mei-P26 and Fused protein levels in *aub* mutant GSCs, and a more prominent increase in Nos levels. Increased Nos protein levels in *aub* mutant GSCs suggest that the direct regulation of *nos* mRNA by Aub occurring in the early embryo is maintained in other biological contexts (Rouget *et al*, 2010).

Cbl protein displayed the highest increased levels in *aub* mutant GSCs compared to control. We thus focused on the possible regulation of the *Cbl* proto-oncogene by Aub. *Cbl* encodes two isoforms through alternative splicing: a long isoform (CblL) and a short isoform (CblS), both of which contain the N-terminal phosphotyrosine binding domain that binds phosphotyrosine kinases, in addition to a ring finger domain that acts as an E3 ubiquitin ligase (Fig 4A; Robertson *et al*, 2000). We used two available monoclonal antibodies directed against either the long Cbl isoform (8C4) or both isoforms (10F1; Pai *et al*, 2006), to analyze the deregulation of *Cbl* in *aub* mutant GSCs. Cbl protein levels were significantly increased in *aub* mutant GSCs as observed with either antibody, although a stronger effect was revealed with the 8C4 antibody (specific to CblL; Fig 4B–E). These results suggest the regulation of

*CblL* mRNA by Aub in the GSCs and are consistent with the reported mRNA expression of *CblL* in germaria and *CblS* in later stages of oogenesis (Pai *et al*, 2006). Quantification of *CblL* mRNA in germaria, using RT–qPCR, showed that the increased Cbl protein levels in *aub* mutant did not result from increased mRNA levels (Fig EV4A).

We used RNA immunoprecipitation with Aub to confirm the potential regulation of *Cbl* by Aub at the mRNA level. GFP-Aub protein was immunoprecipitated from *UASp-GFP-Aub/nos-Gal4* or wild-type (mock immunoprecipitation) ovaries. Quantification of *Cbl* mRNA by RT–qPCR revealed that it was enriched in GFP-Aub immunoprecipitates as compared to the mock immunoprecipitates (Fig 4F). Consistent with Aub interaction with *Cbl* mRNA, Aub iCLIP experiments in 0- to 2-h embryos have revealed the direct binding of Aub to *Cbl* mRNA (Barckmann *et al*, 2015). In addition, a recent study independently reported the role of Aub in GSC self-renewal and differentiation (Ma *et al*, 2017). GFP-Aub iCLIP experiments in cultured GSCs were performed in this study, and we found statistically significant GFP-Aub crosslinks in *Cbl* mRNA 5′- and 3′UTRs, demonstrating Aub direct binding to *Cbl* mRNA in GSCs (Fig 5A). The other mRNAs identified as potential targets of Aub, *nos, mei-P26,* and *fused* were also significantly crosslinked by GFP-Aub in GSCs (Appendix Fig S2).

We then analyzed the potential deregulation of *Cbl* mRNA in the absence of CCR4 deadenylase. We performed Cbl immunostaining using the 8C4 antibody, in *twin* mutant ovaries, as well as in ovaries containing clonal *twin* mutant GSCs. Similar to the results observed in *aub* mutant GSCs, the levels of CblL protein were increased in *twin* mutant GSCs in both experimental conditions (Fig EV4C, C', E–F'', J and K). To address whether the regulation of *Cbl* mRNA by Aub and CCR4 occurred at the level of poly(A) tail length, we measured the poly(A) tail of *CblL* mRNA in early ovaries using ePAT assays. ePAT assays from *bam*[Δ86] ovaries that only contained undifferentiated GSC-like cells confirmed the presence of the long *CblL* mRNA in these cells (Fig 4G). *CblL* poly(A) tails were not notably affected in *twin* and *aub* mutant early ovaries as compared to wild-type (Fig 4G), whereas *mei-P26* used as a control mRNA undergoing deadenylation by CCR4 (Joly *et al*, 2013) had longer poly(A) tails in *twin* mutant (Fig EV4B). This suggested that *CblL* mRNA regulation by Aub/CCR4-NOT did not involve deadenylation. Indeed, the CCR4-NOT complex has the capacity to repress mRNA translation independently of its role in deadenylation, through the recruitment of translational repressors (Chekulaeva *et al*, 2011; Chen *et al*, 2014; Mathys *et al*, 2014).

Taken together, these results reveal *Cbl* mRNA as a direct target of Aub/CCR4-NOT-dependent translational repression in GSCs.

### piRNAs are involved in Cbl mRNA regulation by Aub

We used embryonic (germline) and ovarian (somatic and germline) piRNA libraries to identify piRNAs complementary to *Cbl* mRNA. Strikingly, using strong complementarities (0–3 mismatches, or 20-nt seed/16-nt seed without mismatch in the seed, Barckmann *et al*, 2015), transposable element piRNA target sites were found overlapping significant Aub crosslinks identified in GSCs, in *CblL* mRNA 5′- and 3′UTRs (Fig 5B and C). Mapping of complementary piRNAs to the entire length of *Cbl* mRNAs identified discrete peaks overlapping significant crosslinks in *CblL* (Fig 5D). These peaks were reduced,

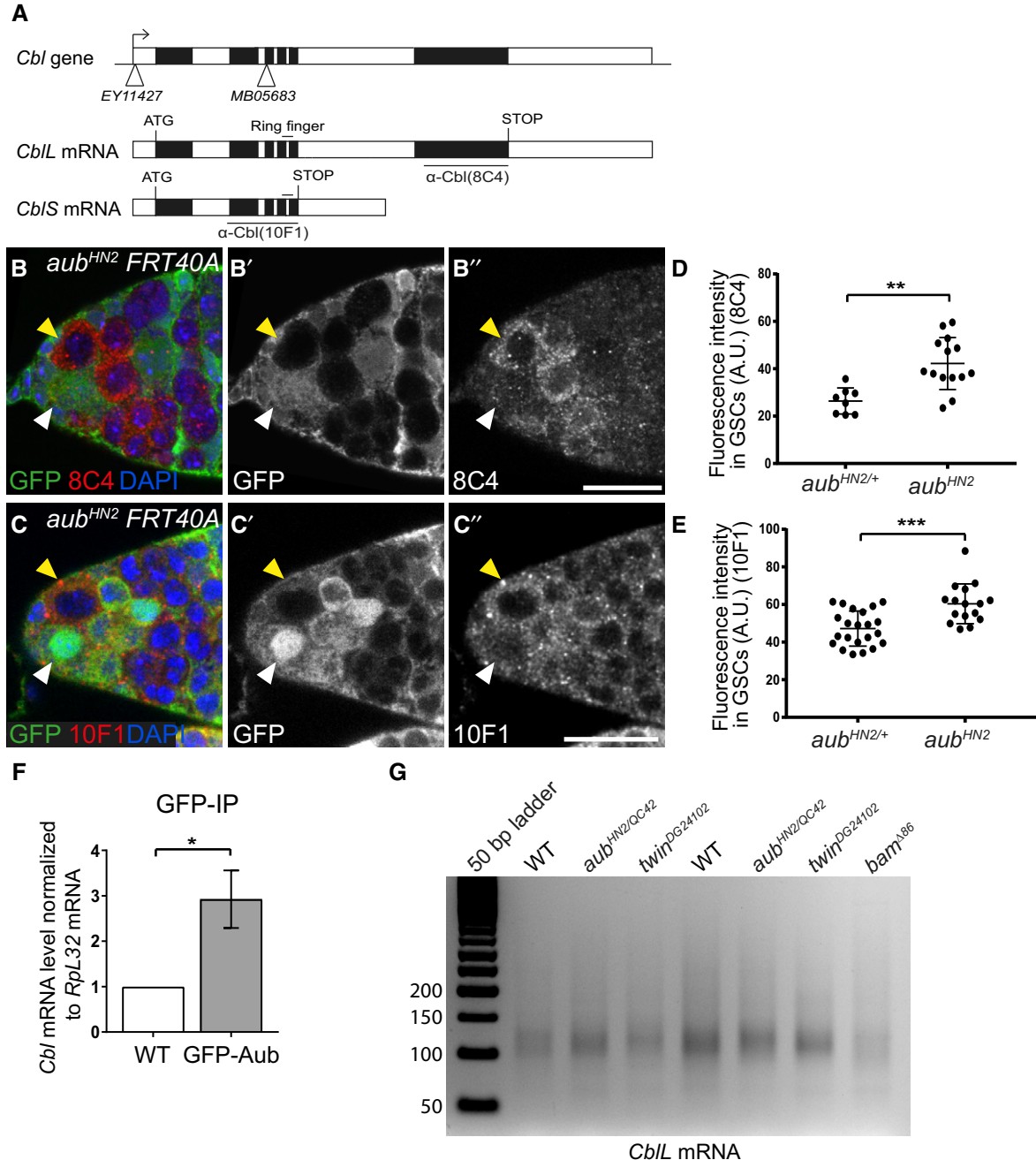

**Figure 4.  Aub represses *Cbl* expression in the GSCs.**

A   Genomic organization of the *Cbl* locus and *Cbl* mRNAs. Open boxes represent UTRs and introns, and black boxes are exons. The insertion points in the *Cbl^EY11427^* and *Cbl^MB05683^* mutants are represented by white triangles. The region encoding the E3 ubiquitin ligase domain (Ring finger) is indicated. The regions used to raise the 8C4 and 10F1 monoclonal antibodies are underlined.

B–C"   Immunostaining of mosaic germaria with anti-GFP (green), to identify clonal cells by the lack of GFP, and either 8C4 (B–B") or 10F1 (C–C") monoclonal anti-Cbl (red). DAPI was used to visualize DNA. White arrowheads indicate *aub^HN2/+^* control GSCs; yellow arrowheads indicate clonal mutant *aub^HN2^* GSCs. Scale bars: 10 μm.

D, E   Quantification of Cbl protein levels in *aub^HN2/+^* and *aub^HN2^* mutant GSCs using fluorescence intensity of immunostaining with 8C4 or 10F1. Fluorescence intensity was measured in arbitrary units using the ImageJ software. Horizontal bars correspond to the mean and standard deviation. \*\*P-value < 0.01, \*\*\*P-value < 0.001 using the two-tailed Student's *t*-test. The number of cells analyzed is indicated as the dots in the figure itself.

F   RNA immunoprecipitation (IP) with anti-GFP antibody in wild-type (mock IP) and *nos-Gal4/UASp-GFP-Aub* ovarian extracts. *Cbl* mRNA was quantified using RT–qPCR. Normalization was with *RpL32* mRNA. Mean of three biological replicates. The error bar represents standard error to the mean. \*P-value < 0.05 using the two-tailed Student's *t*-test.

G   ePAT assay of *CblL* mRNA. Ovaries from 1-day-old (germarium to stage 8) wild-type, *aub* and *twin* mutant females, and from 4- to 7-day-old *bam* mutant females were used.

Source data are available online for this figure.

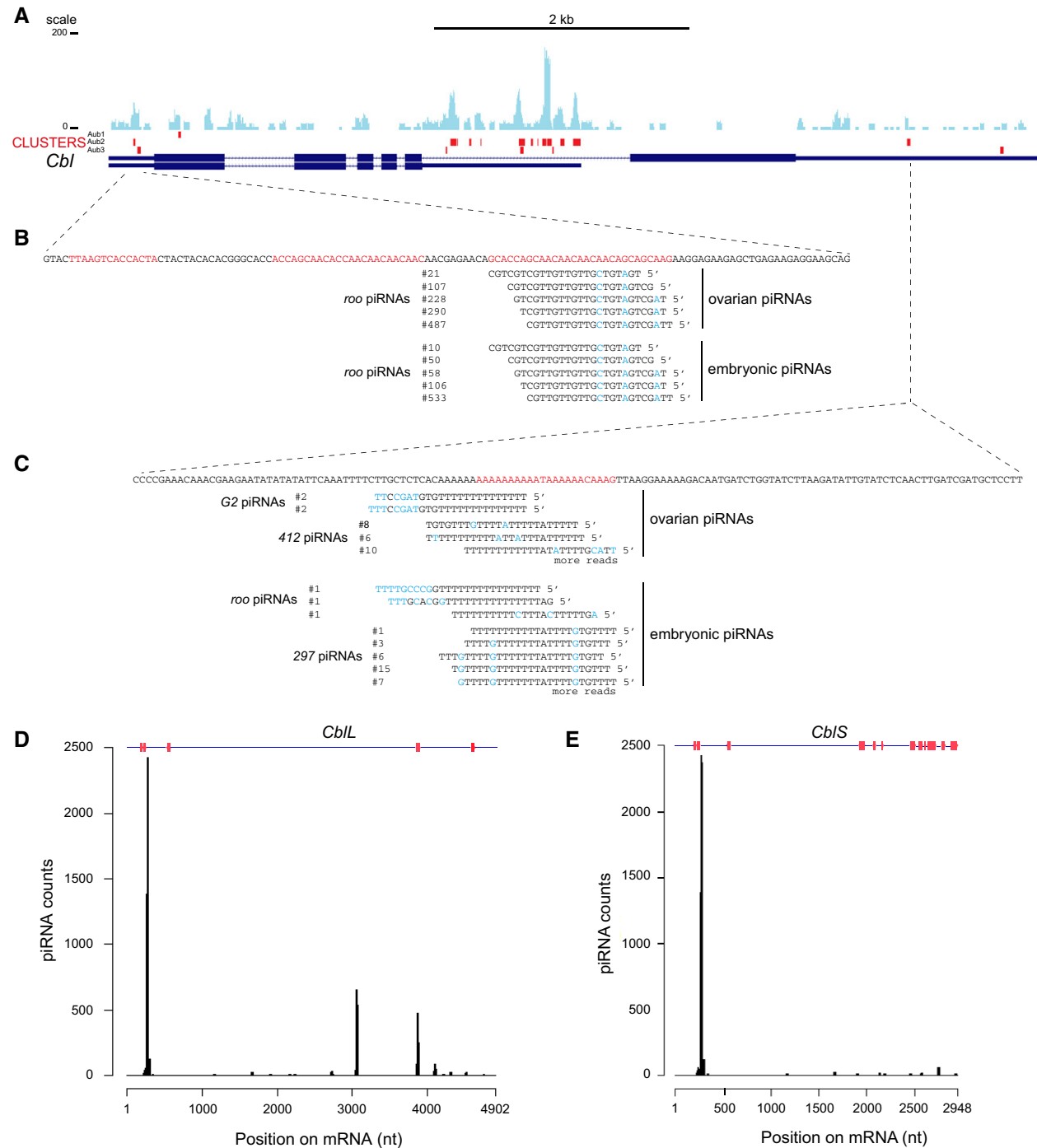

**Figure 5.  GFP-Aub iCLIP in GSCs and targeting by piRNAs, in *Cbl* mRNAs.**

A    GFP-Aub crosslinks in cultured GSCs (Ma *et al*, 2017), in *Cbl* mRNAs. Thin boxes are 5′- and 3′UTRs, lines are introns, and thick boxes are exons. The coverage of *Cbl* mRNAs with iCLIP reads is shown in blue. Significant crosslink clusters are shown in red.

B, C    Sequences of crosslinked regions in 5′UTR (B) and 3′UTR (C) of *CblL* mRNA. The nucleotides (nt) in significant crosslink clusters are in red. The sequence, occurrence, and origin of anti-sense piRNAs from ovarian and embryonic libraries are indicated. Mismatched nt are in blue.

D, E    Coverage of *CblL* (D) and *CblS* (E) mRNAs with ovarian and embryonic piRNAs. Significant crosslink clusters are indicated in red at the top of the graphs.

however, in *CblS* 3′UTR that was heavily bound by Aub (Fig 5E), possibly due to a different mode of Aub binding in this region (e.g., involving reduced piRNA base-pairing and/or additional RNA binding proteins).

To functionally address the role of piRNAs in the regulation of *Cbl* mRNA by Aub, we analyzed Cbl protein levels in GSCs from *armi* mutant females, in which piRNA biogenesis is strongly affected (Malone *et al*, 2009). Immunostaining of *armi* mutant germaria with

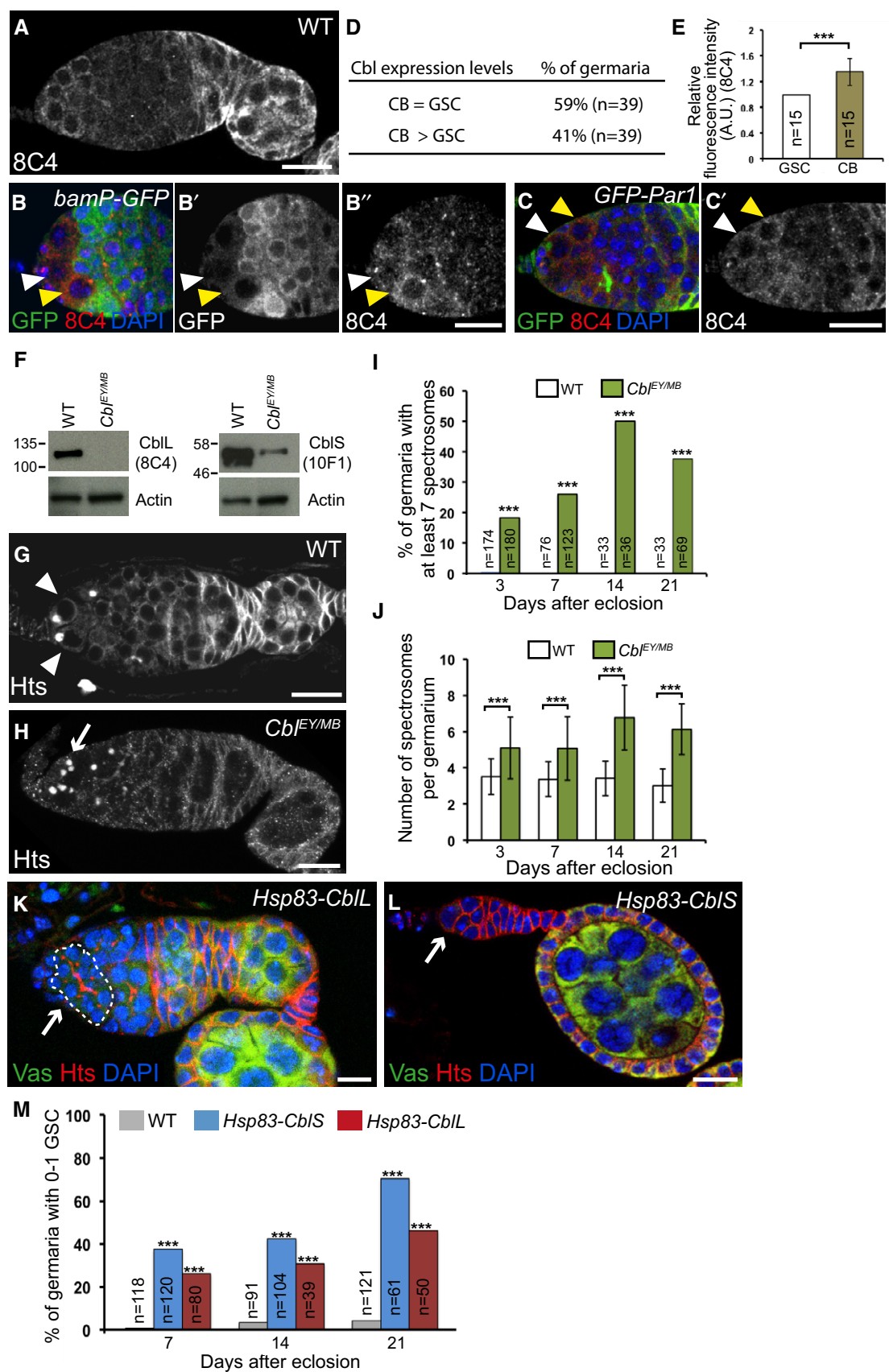

**Figure 6.**

Figure 6. *Cbl* is required for GSC differentiation.

A–E    *Cbl* expression in germaria. (A) Immunostaining of wild-type germaria with anti-Cbl 8C4. (B–B″) Immunostaining of *bamP-GFP* germaria that express GFP under the *bam* promoter, with anti-GFP (green) and anti-Cbl 8C4 (red). (C–C′) Immunostaining of germaria expressing GFP-Par1 to label spectrosomes with anti-GFP (green) and anti-Cbl 8C4 (red). DAPI (blue) was used to visualize DNA. White arrowheads indicate GSCs; yellow arrowheads indicate cystoblasts. (D) Quantification of germaria showing similar or increased Cbl protein levels in cystoblasts as compared to GSCs. (E) Quantification of increased Cbl protein levels in cystoblasts using fluorescence intensity of immunostaining with 8C4 antibody. Fluorescence intensity was measured in arbitrary units using ImageJ, in germaria showing increased Cbl levels in cystoblasts. The number of scored cells (*n*) is indicated. Intensity in GSCs was set to 1. The error bar represents standard deviation. \*\*\**P*-value < 0.001 using the two-tailed Student's *t*-test.
F    Western blots of protein extracts from wild-type and *Cbl* mutant ovaries showing CblL (left panel) and CblS (right panel) revealed with 8C4 and 10F1 antibodies, respectively. A low level of CblS remained, while no CblL was present in the *Cbl^{EY11427/MB05683}* combination. The 10F1 antibody recognizes very poorly CblL in western blot.
G–J    *Cbl* is required for germline differentiation. Immunostaining of wild-type and *Cbl^{EY11427/MB05683}* germaria with anti-Hts to visualize spectrosomes and fusomes. White arrowheads indicate GSCs; the white arrow indicates increased number of spectrosomes. (I) Quantification of germaria with increased number of spectrosomes, and (J) quantification of spectrosomes per germarium. The number of scored germaria (*n*) is indicated in (I). Error bars represent standard deviation. \*\*\**P*-value < 0.001 using the χ² test in (I), and the two-tailed Student's *t*-test in (J).
K–M    *Cbl* induces GSC differentiation. Immunostaining of germaria overexpressing *Cbl* with *Hsp83-CblL* (K) or *Hsp83-CblS* (L) with anti-Vasa (green) and anti-Hts (red). DNA (blue) was revealed with DAPI. White arrows indicate a cyst in the GSC niche (K, outlined) or the loss of GSCs and germ cells (L). (M) Quantification of germaria with 0–1 GSC in the indicated genotypes. The number of scored germaria (*n*) is indicated. \*\*\**P*-value < 0.001 using the χ² test.

Data information: Scale bars: 10 μm in (A–C′), (G), (H), (K), and (L).
Source data are available online for this figure.

anti-Cbl antibody 8C4 revealed a significant increase in CblL levels in *armi* mutant GSCs (Fig EV4G–H′ and L). In contrast, CblL levels were not increased in GSCs from *ago3* mutant, in which piRNAs were produced, and which did not display GSC loss (Fig EV4C–D′ and I).

We conclude that piRNAs that target *CblL* mRNA by complementarity guide Aub interaction with *Cbl*.

## Regulation of *Cbl* mRNA by Aub is required for GSC self-renewal

To determine whether translational repression of *Cbl* mRNA by Aub is relevant to GSC self-renewal, we analyzed the function of *Cbl* in GSCs. We first determined the expression pattern of Cbl in the germarium using immunostaining. The *bamP-GFP* reporter was used to mark the cystoblast to 8-cell cysts (Chen & McKearin, 2003). Co-staining of both Cbl antibodies with GFP revealed the presence of Cbl at low levels in GSCs, cystoblasts, and early (2-cell) cysts. This was followed by reduced levels in the remaining dividing and differentiating cysts up to region 2a of the germarium, and stronger expression starting in region 2b in both somatic and germ cells (Fig 6A–B″). Strikingly, co-staining with anti-Cbl and anti-GFP antibodies of germaria expressing *GFP-Par1* to visualize spectrosomes revealed a transient increase (visible in ≈40% of germaria) of Cbl protein levels in cystoblasts as compared to GSCs (Fig 6C–E, Appendix Fig S3A and B).

Null alleles of *Cbl* are larval/pupal lethal (Pai *et al*, 2000, 2006). To address a potential role for *Cbl* in GSC biology in adults, we used two *Cbl* insertion alleles: *Cbl^{EY11427}*, which contains a *P-UAS* insertion in *Cbl* 5′UTR (Bellen *et al*, 2004), and *Cbl^{MB05683}*, which contains a *Minos*-based insertion in the third exon (Metaxakis *et al*, 2005; Fig 4A). *Cbl^{EY11427/MB05683}* transheterozygotes mostly died at the pupal stage, with a small number of adult escapers surviving for 3–4 days at 25°C. The number of escapers and their survival time increased at 22°C. Western blots of ovaries from these escapers revealed that the CblL isoform was absent in this mutant combination and that CblS levels were strongly reduced (Fig 6F). Anti-Hts immunostaining of ovaries from 3-, 7-, 14-, and 21-day-old *Cbl^{EY11427/MB05683}* mutant females revealed a defect in GSC differentiation. The number of undifferentiated germ cells containing a

spectrosome in *Cbl* mutant germaria was higher than in the wild-type and increased with time (two to four spectrosomes in the wild-type versus six to nine in *Cbl* mutant; Fig 6G–J).

In a reverse experiment, we overexpressed the long or short isoforms of Cbl in germ cells using the *Hsp83-CblL* and *Hsp83-CblS* transgenes (Pai *et al*, 2006) and analyzed germaria using immunostaining with anti-Hts and anti-Vasa. Consistent with a role for *Cbl* in GSC differentiation, overexpressing *Cbl* in the germline led to a GSC loss that increased with time (Fig 6K–M). Germaria with a lower number of germ cells were also visible (Appendix Fig S3C and D). These results identify a new *Cbl* function in GSC homeostasis.

Finally, we addressed whether the translational repression of *Cbl* mRNA by Aub in the GSCs has a functional role in their self-renewal. If increased Cbl protein levels in *aub* mutant ovaries contribute to the *aub* GSC loss phenotype, we would expect to reduce this phenotype by reducing *Cbl* gene dosage. We used both *Cbl^{EY11427}* and *Cbl^{MB05683}* heterozygous mutants in combination with *aub^{HN2/QC42}* and examined the germaria of these females at three time points by immunostaining with anti-Hts and anti-Vasa. Strikingly, the GSC loss phenotype in *aub* mutants was significantly rescued in the presence of both *Cbl* heterozygous mutants (Fig 7A–C). These results show that the regulation of *Cbl* by Aub is essential for GSC self-renewal.

Together, these data identify a new role for *Cbl* in GSC differentiation and reveal an essential role of *Cbl* mRNA translational repression by Aub for GSC self-renewal.

# Discussion

Here, we have demonstrated that *aub* is required intrinsically in GSCs for their self-renewal. The main phenotype of *aub* mutant GSCs is a reduced capacity to self-renew, leading to their progressive loss by differentiation. Our results show that this *aub* function depends only in part on activation of the DNA damage response mediated by Chk2 kinase and involves the regulation of protein-coding genes at the mRNA level.

We provide evidence that the Aub mechanism of action involves the recruitment of the CCR4-NOT deadenylation complex (Fig 7D).

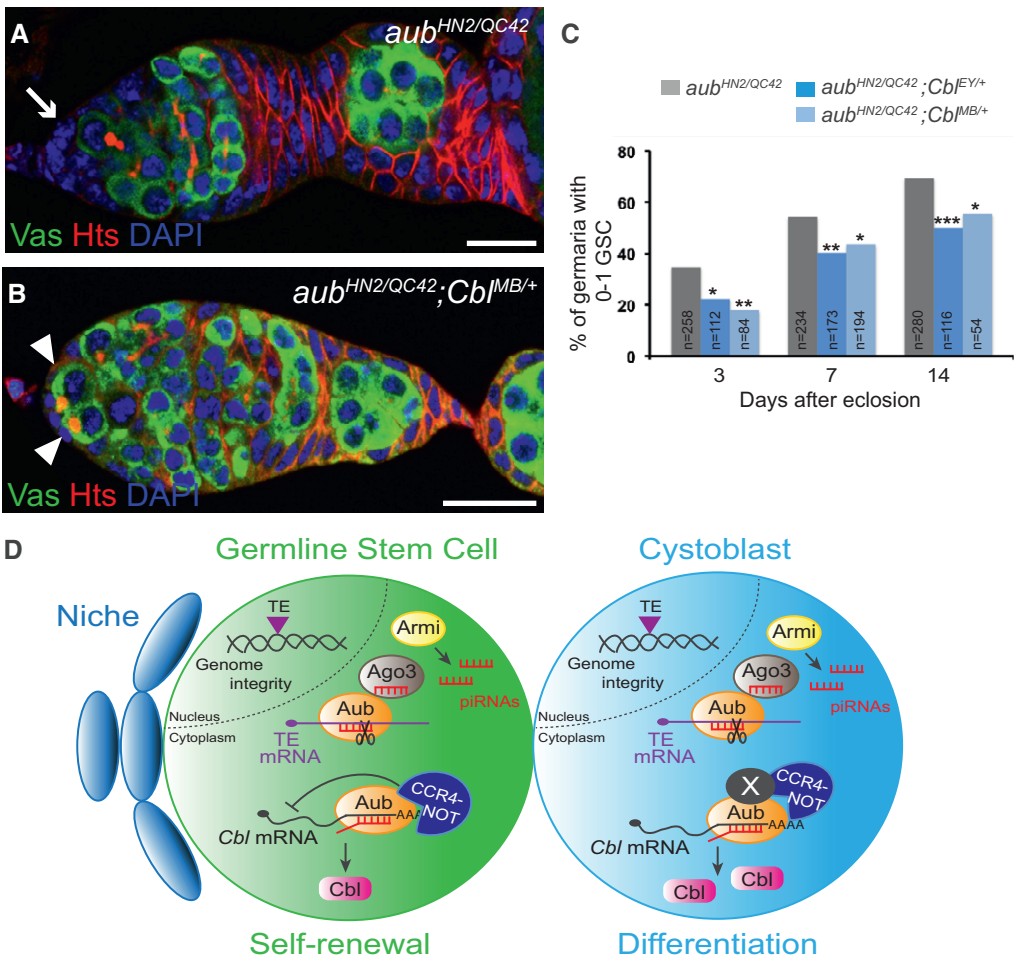

**Figure 7.  Regulation of *Cbl* by Aub in the GSCs is essential for their self-renewal.**

A, B   Immunostaining of germaria from *aub*[HN2/QC42] (A) and *aub*[HN2/QC42]; *Cbl*[MB/+] (B) females with anti-Vasa (green) and anti-Hts (red). DAPI (blue) was used to visualize DNA. The white arrow indicates the lack of GSCs; white arrowheads indicate GSCs. Scale bars: 10 μm.

C   Quantification of mutant germaria with 0–1 GSC in the indicated genotypes. The number of scored germaria (*n*) is indicated. \*\*\**P*-value < 0.001, \*\**P*-value < 0.01, \**P*-value < 0.05, using the $\chi^2$ test.

D   Model of Aub function in GSCs. Aub is required intrinsically in GSCs for their self-renewal and differentiation. Aub function in self-renewal depends on translational repression of *Cbl* mRNA in GSCs through the recruitment of the CCR4-NOT complex. In cystoblasts, this translational repression is decreased, likely through the implication of at least another factor (*X*). As is the case for other translational controls in the GSC lineage, Aub/CCR4-NOT acts in fine-tuning Cbl levels. Aub function in GSC differentiation depends on activation of the Chk2 DNA damage checkpoint, consistent with a role in transposable element (TE) repression to maintain genome integrity; Ago3 has the same role in GSC differentiation.

Aub and subunits of CCR4-NOT, NOT1, NOT3, and CCR4 form a complex in ovaries. Moreover, *aub* heterozygous mutants significantly increase the GSC loss phenotype of a *twin* hypomorphic mutant. Translational repression plays an essential role in GSC biology, for both GSC self-renewal and differentiation. Through their interaction with CCR4-NOT, the translational repressors Nos and Pum repress the translation of differentiation factors in GSCs for their self-renewal (Joly *et al*, 2013); in turn, Pum and Brat recruit CCR4-NOT in the cystoblasts for their differentiation by repressing the translation of self-renewal factors (Harris *et al*, 2011; Newton *et al*, 2015). Therefore, the CCR4-NOT complex is central to mRNA regulation for GSC homeostasis. We have identified Aub as a novel interactor of the CCR4-NOT deadenylation complex involved in translational repression of *Cbl* mRNA for GSC self-renewal. Interestingly, repression of *Cbl* mRNA by Aub/CCR4-NOT does not involve poly(A) tail shortening. This is consistent with the reported role of the CCR4-NOT complex in translational repression, independent of deadenylation. In this mode of regulation, the CCR4-NOT complex serves as a platform to recruit translational repressors such as DDX6/Me31B and Cup (Igreja & Izaurralde, 2011; Chen *et al*, 2014; Mathys *et al*, 2014). Whether CCR4-NOT mediates deadenylation or translational repression might depend on the set of RNA binding proteins involved in its recruitment to specific mRNAs. *mei-P26* mRNA to which CCR4-NOT is recruited by Nos and Pum, in addition to Aub, undergoes deadenylation, whereas *Cbl* mRNA does not. In addition, a role in translational regulation has been proposed for two mouse PIWI proteins, MILI and MIWI. MILI associates with the translation factor eIF3A, while both MILI and MIWI associate with the cap-binding complex (Grivna *et al*, 2006; Unhavaithaya *et al*, 2009). Similarly, Aub might also regulate mRNA translation through

direct interaction with translation factors. Aub and translation initiation factors were recently reported to coprecipitate when overexpressed in S2 cells (Ma et al, 2017).

A major point addressed here is the characterization of the role of the piRNA pathway in GSC biology. Until now, the function of Piwi had been thoroughly analyzed in GSCs. Piwi is involved in somatic niche cells for GSC differentiation, as well as in GSCs for their maintenance and differentiation (Cox et al, 1998, 2000; Jin et al, 2013; Ma et al, 2014). Strikingly, the molecular mechanisms underlying the somatic role of Piwi for GSC differentiation are not related to transposable element repression, but gene regulation. Piwi interacts with components of the PRC2 complex, thus limiting PRC2 binding to chromatin and transcriptional repression (Peng et al, 2016). Piwi also represses c-Fos mRNA via its processing into piRNAs (Klein et al, 2016). Cutoff, a piRNA pathway component that is required for piRNA production, is reported to play a role in GSC self-renewal and differentiation, with a partial rescue of the differentiation defects by mutation in the Chk2 kinase (Chen et al, 2007; Pane et al, 2011). Here, we describe the GSC phenotypes in three additional piRNA pathway mutants: aub, ago3, and armi (Fig 7D). Notably, the three mutants display different defects in GSC biology. Specifically, ago3 mutants only have differentiation defects, whereas the most prominent phenotype of aub and armi mutants is GSC loss. This suggests that different molecular pathways affect GSC biology in these mutants. Importantly, the effect of transposition per se on GSC homeostasis has been analyzed using P-element mobilization in PM hybrid dysgenesis crosses and corresponds to defects in GSC differentiation (Rangan et al, 2011). These defects are partially rescued by Chk2 mutation, indicating that they arise following DNA damage. Accordingly, differentiation defects in aub mutant GSCs are almost completely rescued by Chk2 mutation and might result from transposition. In contrast, GSC loss, the main phenotype in aub mutants, is less strongly rescued by the Chk2 mutant, suggesting that it does not only depend on transposable element mobilization or DNA damage. These results are consistent with our identification of Cbl mRNA regulation by Aub for GSC self-renewal and might be explained by both roles of Aub in transposition repression and cellular mRNA regulation for GSC self-renewal.

Aub function in transposable element regulation occurs through direct cleavage of transposable element mRNAs, guided by complementary piRNAs (Brennecke et al, 2007). Thus, the contribution of Aub/CCR4-NOT interaction to transposable element regulation is expected to be minor. Nonetheless, CCR4 was reported to specifically regulate Het-A transposable element (Morgunova et al, 2015). Whether Het-A repression does contribute to CCR4 function in GSC self-renewal, in addition to its major role in mRNA regulation, remains unknown. The twin GSC loss phenotype was reported to be partially rescued upon Chk2 downregulation using RNAi; however, the rescue was not robust and would require validation using mutants (Fu et al, 2015).

Post-transcriptional regulation of cellular mRNAs by Aub and other PIWI proteins involves piRNAs that target mRNAs by complementarity (Rouget et al, 2010; Gou et al, 2014; Barckmann et al, 2015; Goh et al, 2015). Here, we show that: (i) the GSC loss in aub mutant is not rescued with Aub^AA that is unable to bind piRNAs, (ii) germline piRNAs have the potential to target Cbl mRNAs at Aub binding sites, and (iii) Armi, which has an essential function in piRNA biogenesis, is required for Cbl regulation in GSCs. These

results are consistent with a role of piRNAs in the regulation of mRNAs by Aub in GSCs. These findings broaden the developmental functions of Aub and piRNAs as regulators of gene expression in various biological processes, and highlight their key role in developmental transitions.

A recent study reporting the role of Aub in mRNA regulation in GSCs suggested that this Aub function was independent of piRNAs (Ma et al, 2017). This was based on the lack of piRNA target site enrichment in the regions bound by Aub within 3′UTRs of Aub target mRNAs. This conclusion might differ upon utilization of different piRNA complementarities. Strikingly, this study also reported the GSC loss phenotype of a mutant for dunce (dnc), another Aub target mRNA (Ma et al, 2017). This dnc mutation, dnc^ΔpiR1 (called dnc^3′utrΔ1 in Ma et al, 2017) is a CRISPR-based deletion of a piRNA target site (Barckmann et al, 2015). Thus, the GSC loss phenotype of this mutant supports the role of piRNAs in mRNA regulation by Aub in GSCs.

Importantly, our study reveals a new function for Cbl in GSC biology. Overexpression and mutant analysis show the implication of Cbl in GSC differentiation. Cbl is a tumor suppressor gene encoding an E3 ubiquitin ligase that binds and represses receptor tyrosine kinases. In particular, Cbl regulates epidermal growth factor receptor (Egfr) signaling through ubiquitination and degradation of activated Egfr (Mohapatra et al, 2013). In Drosophila, the regulation of Gurken/Egfr signaling by Cbl is involved in dorsoventral patterning during oogenesis (Pai et al, 2000, 2006; Chang et al, 2008). In humans, mutations in Cbl that disrupt E3 ubiquitin ligase activity lead to myeloid neoplasms (Sanada et al, 2009). Cbl plays a key role in hematopoietic stem cell homeostasis, in the maintenance of their quiescence and their long-term self-renewal capacity (An et al, 2015). Our data thus add a biological function for Cbl to yet another stem cell lineage.

Other E3 ubiquitin ligases are known to play important roles in GSC biology. Mei-P26 and Brat, two members of the conserved Trim-NHL family of proteins, contain E3 ubiquitin ligase domains and have roles in stem cell lineages. Mei-P26 in particular is involved in GSC self-renewal and differentiation, and this dual function partly depends on a very tight regulation of its levels in these cells (Neumuller et al, 2008; Li et al, 2012; Joly et al, 2013). Smurf is another E3 ubiquitin ligase that plays a major role in GSC differentiation. Specifically, Smurf associates with the Fused serine/threonine kinase in cystoblasts to degrade the Thickveins receptor and thus repress BMP signaling, the main signaling pathway in the GSC lineage. This mechanism generates a steep gradient of BMP activity between GSCs and cystoblasts (Xia et al, 2010). Cbl might participate in the regulation of Egfr signaling or other pathways in the GSC lineage. Although the role of Egfr in adult GSC biology in the ovary has not yet been addressed, this signaling pathway is involved in regulating the primordial germ cell number in the larval gonad (Gilboa & Lehmann, 2006) and in the GSC mitotic activity in adult males (Parrott et al, 2012).

Our data highlight an important role of Aub in fine-tuning Cbl levels for GSC self-renewal. Intriguingly, a recent study reported a functional link between Aub and dFMR1 (Fragile X mental retardation protein; Bozzetti et al, 2015). dFMR1 was described previously to regulate Cbl mRNA during oogenesis (Epstein et al, 2009) and to play a role in GSC biology (Yang et al, 2007), thus pointing to the

possibility that Aub and dFMR1 might cooperate for *Cbl* regulation. Further studies will be required to address this question.

Translational regulation is known to be central for cell fate choices in adult stem cell lineages. RNA binding proteins and microRNAs have a recognized regulatory function in female GSCs to trigger cell fate changes through cell-specific regulation of mRNA targets. Here, we reveal piRNAs as an additional layer of translational regulators for GSC biology. PIWI proteins and piRNAs are stem cell markers in somatic stem cells of higher organisms, as well as in pluripotent stem cells involved in regeneration in lower organisms (Juliano *et al*, 2011). This function of piRNAs in translational control is likely to be conserved in these stem cell lineages and might play a key role in stem cell homeostasis.

PIWI proteins and piRNAs are upregulated in a number of cancers, and functional studies in *Drosophila* have shown that this upregulation participates in cancer progression (Janic *et al*, 2010; Fagegaltier *et al*, 2016). This suggests that the role of piRNAs in the translational control of cellular mRNA targets might also be crucial to cancer.

# Materials and Methods

### *Drosophila* stocks and genetics

The $w^{1118}$ stock was used as a control. The following mutant alleles and transgenic lines were used: $aub^{HN2}$ and $aub^{QC42}$ (Schupbach & Wieschaus, 1991), $mnk^{P6}$ (Abdu *et al*, 2002), $twin^{DG24102}$ (Joly *et al*, 2013), $armi^{1}$ (Tomari *et al*, 2004), $armi^{72.1}$ (Cook *et al*, 2004), $ago3^{t1}$, $ago3^{t2}$ and $ago3^{t3}$ (Li *et al*, 2009a), *nos-Gal4:VP16* (Rorth, 1998), *UASp-CCR4-HA* (Semotok *et al*, 2005), *UASp-GFP-Aub* (Harris & Macdonald, 2001), *UASp-GFP-Aub*$^{AA}$ (Barckmann *et al*, 2015), $Cbl^{MB05683}$ and $Cbl^{EY11427}$ (Bloomington *Drosophila* Stock Center), *Hsp83-CblL* and *Hsp83-CblS* (Pai *et al*, 2006), $bam^{A86}$ (McKearin & Ohlstein, 1995), *bamP-GFP* (Chen & McKearin, 2003), and *GFP-Par1* (*Pubq-GFP-Par1*, a gift from A. González-Reyes). The recombinant chromosomes $mnk^{P6} aub^{HN2}$ and $mnk^{P6} aub^{QC42}$ (Klattenhoff *et al*, 2007) were used. Adult females were dissected 3, 7, 14, or 21 days after eclosion. To generate mitotic germline clones, the following stocks were used: $hs\text{-}flp^{1112}$, $aub^{HN2}$ *FRT40A* (this study), *ubi-nls-GFP FRT40A*, *FRT40A*, *FRT82B twin*$^{DG24102}$ and *FRT82B ubi-nls-GFP*. Clones were induced in 3-day-old females with two 1-h heat shocks at 37°C per day, separated by an 8-h recovery period at 25°C, during three consecutive days. Ovaries were dissected 7, 14, or 21 days after the final heat shock.

### Immunostaining and image analysis

Ovaries were dissected at room temperature in PBS supplemented with 0.1% Tween-20 (PBT), fixed with 4% paraformaldehyde, blocked with PBS containing 10% BSA for 1 h and incubated in primary antibodies with PBT 1% BSA overnight at 4°C. Primary antibodies were then washed three times with PBT 1% BSA for 10 min at room temperature. Secondary antibodies were diluted in PBT 0.1% BSA and were incubated for 4 h at room temperature. Secondary antibodies were then washed three times in PBT for 10 min. Primary antibodies were used at the following

concentrations: mouse anti-Hts [1B1; Developmental Studies Hybridoma Bank (DSHB), University of Iowa] 1/100; rabbit anti-Vasa (Santa Cruz Biotechnology) 1/1,000; rat anti-Vasa (DSHB) 1/50; rabbit anti-cleaved Caspase 3 (Biolabs) 1/300; rabbit anti-Bam (a gift from D. Chen) 1/2,000; rabbit anti-Aub (ab17724; Abcam) 1/500; mouse anti-Aub (4D10, Gunawardane *et al*, 2007) 1/1500; mouse anti-HA (ascite produced from 12CA5, Joly *et al*, 2013) 1/2,000; rabbit anti-GFP (A6455; Invitrogen) 1/500; mouse anti-Cbl (8C4; DSHB) 1/50; mouse anti-Cbl (10F1; a gift from LM. Pai) 1/300; rabbit anti-Nanos (a gift from A. Nakamura), 1/1,000; rabbit anti-Brat (Betschinger *et al*, 2006) 1/300; rabbit anti-Mei-P26 (Liu *et al*, 2009) 1/100; mouse anti-Fused (22F10; DSHB) 1/100; rabbit anti-Pgc (Hanyu-Nakamura *et al*, 2008) 1/1,000; rabbit anti-Bruno (Sugimura & Lilly, 2006) 1/3,000; rabbit anti-Lola (a gift from E. Giniger, Giniger *et al*, 1994) 1/100. Secondary antibodies (Alexa 488- and Cy3-conjugated; Jackson ImmunoResearch) were used at 1/300. DNA staining was performed using DAPI at 0.5 μg/ml. Images were captured with a Leica SP8 confocal microscope and analyzed using the ImageJ software. Fluorescence intensity was measured with ImageJ software in wild-type, heterozygous, or mutant GSCs. For each GSC, the mean fluorescence intensity was determined using three independent quantifications in three different cytoplasmic regions in the same confocal section; the number of cells analyzed (*n*) is indicated in the bar graphs, or each cell analyzed is shown as a dot or a square on the graphs.

### Coimmunoprecipitations and Western blots

Protein coimmunoprecipitations were performed using 60 ovaries per experiment from $w^{1118}$, *nos-Gal4/UASp-GFP-Aub*, or *nos-Gal4/UASp-GFP-Aub*$^{AA}$ 3-day-old females. Ovaries were homogenized in 600 μl of DXB-150 (25 mM Hepes-KOH pH 6.8, 250 mM sucrose, 1 mM MgCl$_2$, 1 mM DTT, 150 mM NaCl, 0.1% Triton X-100) containing cOmplete™ EDTA-free Protease Inhibitor Cocktail (Roche) and either RNase Inhibitor (0.25 U/μl; Promega) or RNase A (0.1 U/μl; Sigma). 50 μl of Dynabeads Protein G (Invitrogen) was incubated with 15 μl of mouse anti-GFP antibody (3E6; Invitrogen) for 1 h on a wheel at room temperature. Protein extracts were cleared on 30 μl of Dynabeads Protein G previously equilibrated with DXB-150 for 30 min at 4°C. The pre-cleared protein extracts were incubated with Dynabeads Protein G bound to mouse anti-GFP antibody for 3 h at 4°C. The beads were then washed seven times with DXB-150 for 10 min at room temperature. Proteins were eluted in NUPAGE buffer supplemented with 100 mM DTT at 70°C. Western blots were performed as previously reported (Benoit *et al*, 1999) with antibodies used at the following concentrations: rabbit anti-GFP (Invitrogen) 1/1,000; rabbit anti-CCR4 (Temme *et al*, 2004) 1/1,000; rabbit anti-NOT3 (Jeske *et al*, 2006) 1/2,000; mouse anti-NOT1 (Temme *et al*, 2010) 1/250; mouse anti-Cbl (8C4; DSHB) 1/1,000; mouse anti-Cbl (10F1) 1/1,000; and rabbit anti-actin (Sigma) 1/2,500.

### RNA-immunoprecipitations and RT–qPCR

For RNA-immunoprecipitations (RNA-IP), protein extracts were performed using 300 ovaries from $w^{1118}$ or *nos-Gal4/UASp-GFP-Aub* females. Ovaries were homogenized in 600 μl of DXB-150

containing cOmplete™ EDTA-free Protease Inhibitor Cocktail (Roche) and RNase Inhibitor (0.25 U/μl; Promega). 50 μl of Protein G Mag Sepharose (GE Healthcare) was incubated with 2 μg mouse anti-GFP antibody (3E6; Invitrogen) for 3 h on a wheel at 4°C. Protein extracts were cleared on 50 μl of Protein G Mag Sepharose previously equilibrated with DXB-150 for 30 min at 4°C. The pre-cleared protein extracts were incubated with Protein G Mag Sepharose bound to mouse anti-GFP antibody for 3 h at 4°C. The beads were then washed seven times with DXB-150 for 10 min at room temperature. RNA was prepared using TRIzol (Invitrogen), followed by DNA removal with TURBO DNA-free (Ambion). The total RNA amount was used for reverse transcription; RT–qPCR was performed with the LightCycler System (Roche Molecular Biochemical) using three independent RNA extractions. RT–qPCR to quantify *CblL* mRNA levels was performed as for the RNA-IP, except that 1 μg RNA was used for reverse transcription and RNA was prepared from germaria/early egg chambers dissected from 20 ovaries. Primers used for RT–qPCR were as follows.

| | |
|---|---|
| *Cbl* | Forward CGAACTGAAGGCCATATTCC |
| *Cbl* | Reverse TGTGCTGTTACCGAAGTTGC |
| *CblL* | Forward CGTTGTGGACGCTTTCGATC |
| *CblL* | Reverse CGTTGTGGACGCTTTCGATC |
| *RpL32* | Forward CTTCATCCGCCACCAGTC |
| *RpL32* | Reverse CGACGCACTCTGTTGTCG |

## Poly(A) tail assays

ePAT assays were performed as previously described (Chartier *et al*, 2015) using the following primers.

| | |
|---|---|
| *CblL* | CACGTCATGTAACCGAACAAATC |
| *mei-P26* | CCTCTCTCTTTGTTGAAATCACAAAATGG |

## Bioinformatics and statistics

To map iCLIP reads to the genome, PCR amplification was eliminated by merging into one, duplicate sequences that shared the same random barcodes. Mapping of reads from the three GFP-Aub iCLIP replicates (Ma *et al*, 2017) was performed using Bowtie. Reads were mapped allowing unique hits and no mismatches (Bowtie parameters -v 0 -m 1). The sequencing coverage at each nucleotide position was computed using BEDTools. For prediction of piRNA target sites on mRNAs, we used a pool of piRNAs from ovaries sequenced in published libraries [GSM327620, GSM327621, GSM327622, GSM327623, GSM327624 (Brennecke *et al*, 2008); GSM872307 (Zhang *et al*, 2012); GSM548585 (Rozhkov *et al*, 2010); and GSM948741 (Olivieri *et al*, 2012)]. This led to a total of 3,042,979 non-redundant ovarian piRNA sequences. The pool of embryonic piRNA sequences was described previously (Barckmann *et al*, 2015). Bowtie was used with different complementarities to identify piRNA target sites on transcripts, as follows. Bowtie with option "-v 0", "-v 1", "-v 2", or "-v 3" was used to identify piRNAs complementary to mRNAs with up to 0, 1, 2, or 3 mismatch(es), respectively. For complementarities with a seed (16-nt seed, or

20-nt seed without mismatch), we did not use quality values; therefore, the sum of the quality values at all mismatched read positions ($-e/-maqerr$) was set to an arbitrary value of 2,000, which disabled the quality values. Furthermore, $-l$ (length of the seed) and $-n$ (number of mismatches within the seed) were set to different values. The option "–nofw" was used to search only for reverse complementarity between piRNAs and mRNAs. Statistical tests were performed using the Excel, GraphPad Prism, or online $\chi^2$ test (http://www.aly-abbara.com/utilitaires/statistiques/khi_carre.html) softwares.

**Expanded View** for this article is available online.

## Acknowledgements

We are very grateful to D Chen, E Giniger, A Gonzalez-Reyes, J Knoblich, A Nakamura, LM Pai, M Siomi, and P Zamore for their gifts of fly stocks or antibodies. We thank AL Finoux and W Joly for initial work in this study. This work was supported by the CNRS UMR9002, ANR (ANR-2010-BLAN-1201 01 and ANR-15-CE12-0019-01), and FRM ("Equipe FRM 2013 DEQ20130326534"). PRR held a salary from the Labex EpiGenMed/University of Montpellier and from the Fondation ARC. SP held a salary from FRM "Projets Innovants" and "Equipe FRM 2013".

## Author contributions

PR-R and AC performed the experiments and analyzed the data; SP performed bioinformatic analyses; MS analyzed the data; PR-R and MS designed the study and wrote the manuscript; all authors discussed the manuscript.

## Conflict of interest

The authors declare that they have no conflict of interest.

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
