## [Review Process File · The EMBO Journal]

Manuscript EMBO-2017-97259

Aubergine and piRNAs dfca chY[Yfa `]bY'ghYa `W` self-renewal by repressing the proto-oncogene Cbl

Patricia Rojas-Ríos, Aymeric Chartier, Stéphanie Pierson & Martine Simonelig

Corresponding author: Martine Simonelig, Institute of Human Genetics

Review timeline:

Submission date:	03 May 2017
Editorial Correspondence:	08 June 2017
Editorial Decision:	11 June 2017
Revision received:	24 July 2017
Editorial Decision:	15 August 2017
Revision received:	30 August 2017
Accepted:	04 September 2017

Editor: Anne Nielsen

Transaction Report:

Editorial Correspondence

08 June 2017

Thank you again for submitting your manuscript to The EMBO Journal and sorry that it took a few additional days for me to contact you. As you will see below the editorial decision is not straightforward in this case and I therefore wanted to discuss it with my colleagues in the editorial team, as well as with Bernd Pulverer, before contacting you.

Your manuscript has now been seen by three referees and their comments are included below. As you will see from the reports, the referees all appreciate that you demonstrate a role for Aub in GSC maintenance and link this to post-transcriptional regulation of Cbl. However, you will also see that the referees raise a number of both technical and conceptual concerns that would have to be addressed to fully support that conclusion and make them recommend publication here. In addition, several of these point require additional experimental data of a currently unpredictable outcome, which means that this would likely go beyond the minor revision that we discussed as a pre-requisite for us to go ahead with a revision when you submitted the original pre-submission. I want to emphasise that the competing Dev Cell paper is not taken into account for novelty here (this was also pointed out to the referees).

Based on the interest and timeliness of your findings we would in principle still be willing to consider a revised manuscript but given the extensive and open-ended nature of the issues raised by the referees - as well as the existence of the competing paper - I would like to discuss the experiments that could be included in a potential revision before I go on to make an official decision in this case.

I would therefore ask you to take a look at the reports included below and let me know what kind of

data you would be able to include in a potential revision to address the referee concerns (both in terms of controls for conclusiveness and for further functional insight). I would then take that into consideration - and possibly also discuss it with the referees - before we make a final decision on your study. The aim of this is ultimately to prevent you from working extensively on a revision that would have little chance of convincing the referees. In light of the competitive situation, I would also appreciate a time-estimate for a potential revision.

You can send me the outline for a possible revision (or a preliminary point-by-point response) and I will then get back to you with a decision.

1st Editorial Decision

11 June 2017

Thank you for submitting your manuscript for consideration by The EMBO Journal and for providing me with an outline of the experiments you could include to address the referee concerns.

Given the referees' overall positive recommendations, I would like to invite you to submit a revised version of the manuscript, addressing the comments of all three reviewers along the lines we discussed the other day. I should add that it is EMBO Journal policy to allow only a single round of revision, and acceptance of your manuscript will therefore depend on the completeness of your responses in this revised version.

Thank you for the opportunity to consider your work for publication. I look forward to your revision.

REFEREE REPORTS

Referee #1:

In this manuscript, Rojas-Rios and colleagues report the function of the gene aubergine (*aub*) in the maintenance of germline stem cells in *Drosophila*. *Aub* is well known for its functions both in localizing mRNA during *Drosophila* oogenesis, and in producing piRNAs, which silence transposable elements. Simonelig and colleagues have proposed previously that both functions could be linked and that *Aubergine* could repress mRNA translation through its binding to piRNAs (Rouget et al. 2010; Barckmann et al., 2015). Here, they propose a similar model for *Aub* function in GSC, whereby it would inhibit *Cbl* mRNA translation in GSC through its binding to piRNAs. Accordingly, overexpression of *cbl* seems sufficient to induce GSC differentiation. Overall, the experiments are well-performed and controlled. The results are convincing, with a few exceptions (see below). My main criticism relate to how the authors fit their data with the existing literature. Several of the genetic and biochemical interactions have been published previously and are not cited correctly (role of *Aub*, *Cbl*, *Twin* in GSC maintenance, etc...). A few experiments would help integrate their results with published data or at least it should be discussed in the manuscript. Nonetheless, the authors have identified *cbl* mRNA as a target, and genetic interaction with *cbl* mutant support their model. Finding at least one target that makes sense in this kind of studies is always an achievement, which has to be acknowledged.

Main comments:

- 1) *Aub* in GSCs: an identical study on *aub* function in GSC has been published very recently by Ting Xie's lab (Ma et al., *Developmental Cell*, 2017). Surprisingly, the results and model proposed are most of the time opposite to the results shown here: *Aub* would increase translation and without involving piRNAs, DNA damages cause GSC loss, which then can be rescued by checkpoint silencing, etc...I am not asking the authors to explain such differences, but at least to discuss how identical experiments can lead to such opposite results.
- 2) *Cbl* and GSC: the authors identified *cbl* mRNA as the main target of *Aubergine* for GSC self-renewal. However, they fail to mention previous studies on the role of *cbl* and the Fragile X protein (*dFMR1*) in the maintenance of GSC. *dFMR1* was shown to be required for GSC maintenance

(Yang et al., 2007) and to inhibit *cbl* mRNA during *Drosophila* oogenesis (Epstein et al., 2009). Furthermore, *dFMR1* was shown to interact biochemically and genetically with Aubergine and piRNAs during oogenesis. It seems important to characterize the link between *dFMR1*, Aub and Cbl, and the authors' model in light of previously published reports.

3) Twin and GSC maintenance: Fu et al. 2015 have already reported by coIP that Twin is in a complex with Aubergine (and Ago3). They further showed that Twin is required for TE silencing and GSC maintenance. Interestingly, the loss of GSCs can be rescued by removing *chk2* activity. These data should be integrated in the authors' study and at least cited.

4) The data showing higher expression of *cbl* mRNA in cystoblast vs GSC is not convincing (Figure 5). It does not seem to be required for the authors' model. Is there a lower expression of Aubergine in cystoblasts?

Specific comments:

1) Ago3 does not seem to play a role in the GSC self-renewal (Figure S2). Ago3 and Aub are well-known partners in producing germinal piRNAs, it would rather suggest that Aub function in GSC is independent of piRNAs. Ago3 is also known to be required (directly or indirectly through piRNAs) for Aub localization. How is Aub localization in GSC mutant for Ago3?

2) The experiment of Fig. 4G ePAT assays is not clear. In the line corresponding to *bam* mutant no band is visible whereas in the text is written that "...confirms the presence of the long CblL mRNA in these cells..."

Referee #2:

The article entitled "Aubergine and piRNAs repress the proto-oncogene *Cbl* for germline stem cell self-renewal" has addressed the mode of actions of Aub in germline stem cell (GSC) biology. The functions of Aub in GSC have not been well studied thus far. This study shows that Aub and Ago3 together act in GSC differentiation, while Aub plays a unique role on GSC maintenance in a manner independent of transposon repression and the DNA damage response pathway, but through regulation on *Cbl*. This study shows *Cbl*, a known target of Aub in embryo, is required for proper GSC differentiation. *Cbl* expression is repressed in GSC by Aub-piRNA complex that is physically and genetically interacting with CCR4-NOT. *Cbl* has been characterized in hematopoietic stem cell quiescence/exhaustion, and the provided findings can broaden the view in which *Cbl* is involved in stem cell biology. However, this study lacks the depth of analyses for some important aspects, and does not provide the novelty expected for articles in EMBO Journal. Though the presented evidence supports the author's claims to some extent, this article would be more suitable in other journals oriented towards germ/stem cell development. The following points are to be considered.

Major points

1) The lack of evidence showing specific function of Aub in GSC.

Critically, it is not clear how Aub and the bound piRNA can repress *Cbl* specifically in GSC but not in the differentiating cyst cells. Their previous study and the presented data in this study indicate that interaction between Aub-piRNA and *Cbl* is maintained through oogenesis and in embryo (as they used entire 1-day old ovary for IP containing germarium and young egg chambers). In addition, though it is not described, CCR4-HA seems to recruit Aub outside of GSCs as well (Fig 3B).

How can Aub repress *Cbl* expression specifically in GSC but not in cyst? Obviously Aub expression appears to stay in a similar level in GSC and cysts. Is there any piRNA specifically expressed in GSC and targeting *Cbl* mRNA (refer to point 2)?

As to Aub specificity, *armi* mutant also significantly increases the level of *Cbl* in GSC (Fig S3GH; much more than *aub* mutant). As mentioned in this study, there are series of studies on Piwi in GSC maintenance and differentiation. Is Piwi involved in the upregulation of *Cbl* and GSC loss? How about Ago3, the other PIWI family protein involved in secondary piRNA amplification in germ cells? The authors should think of having Ago3 throughout the study as a control, to convince specific function of Aub for GSC maintenance.

2) The lack of molecular mechanism how Aub-bound piRNA represses *Cbl*.

Previously the authors reported Aub-bound piRNA in embryo and indeed they used the data to

screen potential candidates which are repressed by Aub-piRNA in GSC. I wonder if complementary piRNAs to cbl mRNA can be found.

Related to this point, in Fig 2, the authors showed that Aub-AA mutant which does not bind piRNA failed in complementing aub mutant phenotype. However, since the authors did not even examine if the AA mutation affects CCR4-NOT interaction, they cannot conclusively say that piRNA is involved in Cbl repression in GSC.

3) The authors claim that Aub/CCR4-NOT repress the translation of the Cbl mRNA. However, the mRNA level was not quantified. qRT-PCR or in situ hybridization should be conducted.

4) In Fig 4F, can GFP-Aub still interact with Cbl mRNA in the twin mutant background? Are Mei-P26, Fused, and Nanos (analyzed in Aub clonal analysis) mRNAs co-IPed with GFP-Aub as Cbl mRNA? Related to the point 1, to show the GSC-specific control of Cbl by Aub, the authors can also try GFP-IP in bam mutant background (GSC tumor).

5) In Fig 4G, for clear conclusion for Cbl mRNA deadenylation-independent regulation, it is required to include a reference mRNA which is degraded by deadenylation (and the cancellation by twin mutant). bam[delta86] had much less signal. Does this mean Cbl mRNA is low in GSC tumor? The control gene is needed for this analysis. In addition, to show the deadenylation event strictly in GSC, aub or twin mutation should be combined with bam mutant (GSC tumor). How can the authors identify A12 position? The size ladder is needed.

6) Fig S5, 5A-D; the authors tested immunostaining of Cbl in twin mutant. The effect was much weaker than Aub mutant clone. The reduced Cbl level is not evident in WT GSC compared to CB. This is inconsistent with Fig. 5D. Additionally, I do not see the difference of 8C4 signal between GSC and CB in Fig 5A. Fig 5D needs more sample number (it is currently only 8) like other analyses.

7) Fig 6C and Fig 3G; can cbl[EY] or cbl[MB] mutation rescue the GSC loss phenotype in aged twin mutant? (twin[DG24102]/cbl[EY or MB]/ twin[DG24102])

Minor points

1) In Fig1E, does the green color show GFP-Aub or Vasa (the legend says Vasa for green)?

2) Description format of the transheterozygous genotype is not unified, like aub[HN2/QC42] versus aub[HN2]/aub[QC42].

Referee #3:

The manuscript by Rojas-Rios et al. makes a substantial contribution to our understanding of *Drosophila* germline stem cell (GSC) biology. It identifies Cbl as a novel factor that promotes GSC differentiation. Furthermore, it defines a role for Aubergine (Aub) in GSC maintenance, which it carries out by repressing Cbl translation by recruiting the CCR4-NOT to the Cbl 3' UTR.

In general I find the experiments are carefully performed and convincingly support the conclusions the authors wish to draw. I do however have a few specific concerns.

1. Most importantly, while substantial evidence is presented that piRNAs in general are required for Aub-mediated regulation of Cbl, no hypothesis is presented as to what the specific role of these piRNAs might be. For example, does Cbl mRNA have sites of complementarity with some Aub-associated piRNA, that might contribute to recruitment of Aub? The significance of the work would be greater if this question were better resolved.

2. In Fig 3C the result for NOT1 is very clear but the co-IP of CCR4 with GFP-Aub is not much above the (high) background signal. Can a different CCR4 antibody be tried for this experiment?

3. In Fig 4G I don't understand what is going on with Cbl poly(A) tail length in the bam mutant as

there is very little signal at all. Nor do I understand how this result supports the sentence on page 12 that begins "ePAT assays from bam[delta]86 ovaries..."

4. Also with regard to Fig 4G to my eye the average poly(A) tail length looks longer in the aub lane than in wild-type and in twin. As this is contrary to the conclusion as stated in the text, this experiment should be repeated multiple times and the results quantified.

1st Revision - authors' response

24 July 2017

Dr Martine Simonelig
mRNA Regulation and Development
Institute of Human Genetics, 141, rue de la Cardonille
34396 Montpellier Cedex 5. France
Tel: 33 434359959
e-mail: Martine.Simonelig@igh.cnrs.fr

Dr Anne Nielsen, Editor
The EMBO Journal

Manuscript number: EMBOJ-2017-97259R

Montpellier, July 24, 2017

Dear Dr Nielsen,

I thank you very much for your recent decision regarding our manuscript number EMBOJ-2017-97259, "Aubergine and piRNAs repress the proto-oncogene *Cbl* for germline stem cell self-renewal" by Rojas-Rios et al. We are now submitting a revised version of the manuscript.

We thank the referees for their comments that have been very helpful in strengthening the manuscript. We have addressed all their concerns in this revision, and the manuscript is now expanded by two Figures (Figure 5 and Appendix Figure 2) and additional data in 8 Figures.

The point-by-point Response to the Referees is as follows.

Referee #1:

In this manuscript, Rojas-Rios and colleagues report the function of the gene aubergine (*aub*) in the maintenance of germline stem cells in *Drosophila*. *Aub* is well known for its functions both in localizing mRNA during *Drosophila* oogenesis, and in producing piRNAs, which silence transposable elements. Simonelig and colleagues have proposed previously that both functions could be linked and that Aubergine could repress mRNA translation through its binding to piRNAs (Rouget et al. 2010; Barckmann et al., 2015). Here, they propose a similar model for *Aub* function in GSC, whereby it would inhibit *Cbl* mRNA translation in GSC through its binding to piRNAs. Accordingly, overexpression of *cbl* seems sufficient to induce GSC differentiation. Overall, the experiments are well-performed and controlled. The results are convincing, with a few exceptions (see below). My main criticism relate to how the authors fit their data with the existing literature. Several of the genetic and biochemical interactions have been published previously and are not cited correctly (role of *Aub*, *Cbl*, *Twin* in GSC maintenance, etc...). A few experiments would help integrate their results with published data or at least it should be discussed in the manuscript. Nonetheless, the authors have identified *cbl* mRNA as a target, and genetic interaction with *cbl* mutant support their model. Finding at least one target that makes sense in this kind of studies is always an achievement, which has to be acknowledged.

Main comments:

1) *Aub* in GSCs: an identical study on *aub* function in GSC has been published very recently by Ting Xie's lab (Ma et al., *Developmental Cell*, 2017). Surprisingly, the results and model proposed are most of the time opposite to the results shown here: *Aub* would increase translation and without involving piRNAs, DNA damages cause GSC loss, which then can be rescued by checkpoint silencing, etc...I am not asking the authors to explain such differences, but at least to discuss how identical experiments can lead to such opposite results.

Importantly, the *aub* phenotypes in GSC biology: GSC loss and differentiation defects, are reproduced in both studies. Concerning the rescue of *aub* phenotypes by *Chk2* mutation, we agree

that the results should also corroborate in both studies. We believe that the different interpretations might come from the different methods both labs used to quantify the phenotypes. In our study, we separate the phenotypes of GSC loss and tumors (differentiation defects), whereas Ma et al. do not make that distinction. This could affect, to some extent, the rescue of GSC loss by Chk2 mutant. Another difference was the way we quantified GSC loss in the previous version: we counted germaria with lower numbers of GSCs (0 to 1), whereas Ma et al. counted the number of GSCs per germarium. Prompted by this remark, we have now counted the number of GSCs per germarium as in Ma et al., and we find a partial rescue of *aub* mutant GSC loss with the Chk2 mutant. These new data are now included in Figure 2C and Figure EV1G.

Regarding the role of piRNAs in Aub function, we have now performed two sets of experiments to confirm that Aub function in GSCs is dependent of piRNAs: bioinformatic analyses on Ma et al. data sets of GFP-Aub iCLIPs in cultured GSCs, and new co-IP between Aub^{AA} (Aub mutant unable to load piRNAs) and the CCR4-NOT complex. These experiments were specific requests from Referee #2 (point 2) and Referee #3 (point 1); these new results are shown in Figures 3 and 5. Moreover, Ma et al. also have one piece of information that suggests that piRNAs might be involved. They are using a *dnc* mutant that we have generated in Barckmann et al. (*dnc*^{Δ*piR1*}, Cell Reports, 2015, 12:1205-1216). *dnc*^{Δ*piR1*} is a CRISPR mutant that deletes a piRNA target site and Ma et al. found that it shows GSC loss. This is consistent with a potential role of piRNAs in the regulation of *dnc* by Aub. Note that this mutant was misnamed *dnc*^{3'*UTRA1*} in the Ma et al. publication.

Concerning the mechanism of action of Aub, either repressor or activator, Aub and piRNAs have been described in many studies to repress (not activate) mRNAs of transposons and cellular genes. In GSCs, both studies analyze the regulation of different mRNAs that could be regulated by different mechanisms.

2) *Cbl* and GSC: the authors identified *cbl* mRNA as the main target of Aubergine for GSC self-renewal. However, they fail to mention previous studies on the role of *cbl* and the Fragile X protein (dFMR1) in the maintenance of GSC. dFMR1 was shown to be required for GSC maintenance (Yang et al., 2007) and to inhibit *cbl* mRNA during drosophila oogenesis (Epstein et al., 2009). Furthermore, dFMR1 was shown to interact biochemically and genetically with Aubergine and piRNAs during oogenesis. It seems important to characterize the link between dFMR1, Aub and *Cbl*, and the authors' model in light of previously published reports.

dFmr1 was shown to be required in GSC self-renewal, however this function is non-cell autonomous, and should therefore depend on *dFmr1* function in somatic niche cells (Yang et al., HMG, 2007). In addition, *Cbl* mRNA regulation by *dFmr1* was based on variation of *Cbl* mRNA levels in *dFmr1* mutant ovaries, without an effect at the protein level (Epstein et al., Developmental Biology, 2009). Therefore, the regulation of *Cbl* mRNA by dFMR1 might be independent to its regulation by Aub. Nonetheless, the recently reported link between dFMR1 and Aub (Bozzetti et al., JCS, 2015) is intriguing, and we have introduced this information in the Discussion p. 19.

3) *Twin* and GSC maintenance: Fu et al. 2015 have already reported by coIP that *Twin* is in a complex with Aubergine (and Ago3). They further showed that *Twin* is required for TE silencing and GSC maintenance. Interestingly, the loss of GSCs can be rescued by removing *chk2* activity. These data should be integrated in the authors' study and at least cited.

We previously reported co-IPs between Aub, Ago3 and CCR4 (but not with Piwi) with endogenous proteins in embryos, in the presence of RNase (Rouget et al. Nature, 2010, 467: 1128-1132). We have now included this information in the manuscript on p. 9. Fu et al. (Cell Reports, 2015) showed that *Twin* (CCR4) can co-IP Aub and Ago3 when overexpressed in S2 cells; this does not add to our previous results and is less relevant since overexpressed exogenous proteins were used. We have previously reported that *twin* is required for GSC maintenance (Joly et al., Stem Cell Reports, 2013, 1: 411-424); Fu et al. reported the same result (Cell Reports, 2015). Both references are now indicated on p. 10.

The role of *twin* in TE silencing and Chk2-dependent DNA damage checkpoint activation in Fu et al. (Cell Reports, 2015) is less robust. First, the deregulation of only two transposons was reported, *Tart* and *Het-A*, and the effect on *Tart* was very weak. *Het-A* transposon is specifically

localized at telomeres and involved in telomere maintenance. Another report showed the specific deregulation of *Het-A* (as compared to five other transposable elements, including *Tart*) in *twin-KD* ovaries (Morgunova et al., NAR, 2015), indicating that *Het-A* regulation by CCR4 is specific and does not reflect general transposable element regulation. Second, the experiment showing the rescue of *twin* phenotype by Chk2 KD, in Fu et al. (Cell Reports, 2015, Figure 3F) was performed using RNAi. RNAi was induced at 29°C, a temperature that induces ovarian defects in wild-type flies independently of RNAi (our unpublished data). The right control of the *twin-KD lok-KD* experiment in Figure 3F is *twin-KD gfp-KD* (the same number of *UAS* transgenes with an irrelevant KD), however, the statistics between *twin-KD lok-KD* and *twin-KD gfp-KD* were not calculated and might not have been significant because, the negative control *twin-KD gfp-KD* showed indeed some level of rescue. Therefore, to conclude on the rescue of the *twin* GSC loss phenotype with Chk2, the experiment should be performed using mutants. These information have been introduced in the Discussion p. 17.

4) The data showing higher expression of *cbl* mRNA in cystoblast vs GSC is not convincing (Figure 5). It does not seem to be required for the authors' model. Is there a lower expression of Aubergine in cystoblasts?

We have performed more Cbl staining in wild-type germaria, which confirmed a transient increase in Cbl levels in cystoblast as compared to GSCs. We have now quantified the percentage of cystoblasts showing this increase (Figure 6D), and we have increased the number for quantification of Cbl levels in cystoblasts and GSCs (Figure 6E).

We have performed new Aub staining in wild-type germaria (Figure EV2A). Aub does not appear to be expressed at lower levels in cystoblasts.

Specific comments:

1) Ago3 does not seem to play a role in the GSC self-renewal (Figure S2). Ago3 and Aub are well-known partners in producing germinal piRNAs, it would rather suggest that Aub function in GSC is independent of piRNAs. Ago3 is also known to be required (directly or indirectly through piRNAs) for Aub localization. How is Aub localization in GSC mutant for Ago3?

In the absence of Ago3, Aub/Aub homotypic ping-pong increases and piRNA production is not abolished (Li et al. Cell, 2009); these remaining piRNAs in *ago3* mutant would be involved in Aub-dependent regulation in GSCs. This information is indicated on p. 9.

We have analyzed Aub expression in *ago3* mutant GSCs; Aub is expressed similarly as in wild-type GSCs. This new result is shown in Figure EV2A-B'.

2) The experiment of Fig. 4G ePAT assays is not clear. In the line corresponding to *bam* mutant no band is visible whereas in the text is written that "...confirms the presence of the long Cbl mRNA in these cells..."

This point was also raised by Referee #3 (point 3). We have now provided a new picture in Figure 4G, which also shows several examples of *Cbl* mRNA ePAT in wild-type, *aub* and *twin* mutants. More examples for *bam*, *twin* and *aub* mutants are shown in the Source Data For Figure 4G.

Referee #2:

The article entitled "Aubergine and piRNAs repress the proto-oncogene *Cbl* for germline stem cell self-renewal" has addressed the mode of actions of Aub in germline stem cell (GSC) biology. The functions of Aub in GSC have not been well studied thus far. This study shows that Aub and Ago3 together act in GSC differentiation, while Aub plays a unique role on GSC maintenance in a manner independent of transposon repression and the DNA damage response pathway, but through regulation on *Cbl*. This study shows *Cbl*, a known target of Aub in embryo, is required for proper GSC differentiation. *Cbl* expression is repressed in GSC by Aub-piRNA complex that is physically and genetically interacting with CCR4-NOT. *Cbl* has been characterized in hematopoietic stem cell quiescence/exhaustion, and the provided findings can broaden the view in which *Cbl* is involved in stem cell biology. However, this study lacks the depth of analyses for some important aspects, and does not provide the novelty expected for articles in EMBO Journal. Though the presented evidence supports the author's claims to some extent, this article would be

more suitable in other journals oriented towards germ/stem cell development. The following points are to be considered.

Major points

1) The lack of evidence showing specific function of Aub in GSC.

Critically, it is not clear how Aub and the bound piRNA can repress *Cbl* specifically in GSC but not in the differentiating cyst cells. Their previous study and the presented data in this study indicate that interaction between Aub-piRNA and *Cbl* is maintained through oogenesis and in embryo (as they used entire 1-day old ovary for IP containing germarium and young egg chambers). In addition, though it is not described, CCR4-HA seems to recruit Aub outside of GSCs as well (Fig 3B).

How can Aub repress *Cbl* expression specifically in GSC but not in cyst? Obviously Aub expression appears to stay in a similar level in GSC and cysts. Is there any piRNA specifically expressed in GSC and targeting *Cbl* mRNA (refer to point 2)?

We agree with the referee that Aub-*Cbl* mRNA interaction is likely to be maintained in oogenesis outside GSCs. Although we do not know how the regulation switches in cystoblasts we propose in our model (Figure 7D) the implication of one or several other components. It has been shown that associations between translational regulators evolve from GSCs to cystoblasts and cysts, to trigger stage-specific regulation of mRNA targets and drive differentiation processes. In particular, Pumilio interacts with Nanos and CCR4 in GSCs to repress differentiation mRNAs, whereas it interacts with Brat and CCR4 in cystoblasts to repress self-renewal mRNAs (Harris et al. *Developmental Cell*, 2011; Joly et al. *Stem Cell Reports*, 2013; Newton et al. *Development*, 2015).

Our bioinformatic analyses shows that *Cbl* mRNA is targeted by piRNAs (Figure 5).

However, it is unlikely that piRNAs present in GSCs would not be transferred to cystoblasts, and therefore that different content in piRNAs in the two cell types would be the molecular basis of the differential regulation.

As to Aub specificity, *armi* mutant also significantly increases the level of *Cbl* in GSC (Fig S3GH; much more than *aub* mutant). As mentioned in this study, there are series of studies on Piwi in GSC maintenance and differentiation. Is Piwi involved in the upregulation of *Cbl* and GSC loss? How about Ago3, the other PIWI family protein involved in secondary piRNA amplification in germ cells? The authors should think of having Ago3 throughout the study as a control, to convince specific function of Aub for GSC maintenance.

A role of *Armi* in upregulation of *Cbl* mRNA in GSCs is expected if piRNAs are required for this regulation, since *Armi* is a key component of piRNA biogenesis (Malone et al., *Cell*, 2009). As explained in Specific point 1 from Referee #1, Aub/Aub homotypic ping-pong increases in the absence of Ago3 and the biogenesis of piRNAs is not abolished (Li et al., *Cell*, 2009). We show in Figure EV2 that *ago3* mutant has a very different phenotype than *aub* mutant, since it does not show a GSC loss phenotype, but only a defect in differentiation. Thus, Ago3 is not involved in GSC self-renewal. We have now performed new experiments to record *Cbl* protein levels in *ago3* mutant GSCs; they show that *Cbl* levels are not increased in *ago3* mutant (Figure EV4 D-E).

The role of Piwi in GSC biology has been addressed in many studies and is very different to the regulation by Aub that we address here. Piwi acts in the nucleus, and important Piwi functions for GSC maintenance occur in the somatic niche cells (Jin et al., *Current Biology*, 2013; Ma et al., *PLoS One*, 2014; Peng et al., *Nature Genetics*, 2016). Due to these major differences with *Cbl* mRNA regulation by Aub, occurring at the post-transcriptional level and within GSCs, we have not addressed the role of Piwi in *Cbl* regulation.

2) The lack of molecular mechanism how Aub-bound piRNA represses *Cbl*.

Previously the authors reported Aub-bound piRNA in embryo and indeed they used the data to screen potential candidates which are repressed by Aub-piRNA in GSC. I wonder if complementary piRNAs to *cbl* mRNA can be found.

We have used the data set of GFP-Aub iCLIP in cultured GSCs, published in Ma et al. (*Dev. Cell*, 2017) to identify the sites of Aub interaction in *Cbl* mRNA. We then identified piRNAs complementary to *Cbl* mRNA in ovarian and embryonic piRNA libraries. Strikingly, abundant piRNAs were found in close proximity to Aub crosslink sites. These new data are shown in Figure

5.

Related to this point, in Fig 2, the authors showed that Aub-AA mutant which does not bind piRNA failed in complementing aub mutant phenotype. However, since the authors did not even examine if the AA mutation affects CCR4-NOT interaction, they cannot conclusively say that piRNA is involved in Cbl repression in GSC.

This is a very good suggestion; we performed these experiments and found that Aub^{AA} retains the capacity to interact with the CCR4-NOT complex in co-IPs. These data are shown in Figure 3D.

3) The authors claim that Aub/CCR4-NOT repress the translation of the Cbl mRNA. However, the mRNA level was not quantified. qRT-PCR or in situ hybridization should be conducted.

We have quantified *Cbl* mRNA levels by RT-qPCR in *aub* mutant dissected germaria and found that they are not increased as compared to wild type. This experiment is shown in Figure EV4A.

4) In Fig 4F, can GFP-Aub still interact with Cbl mRNA in the twin mutant background? Are Mei-P26, Fused, and Nanos (analyzed in Aub clonal analysis) mRNAs co-IPed with GFP-Aub as Cbl mRNA? Related to the point 1, to show the GSC-specific control of Cbl by Aub, the authors can also try GFP-IP in *bam* mutant background (GSC tumor).

The CCR4 deadenylase does not bind mRNAs specifically; it is recruited to mRNAs by RNA binding proteins. Using Ma et al. GFP-Aub iCLIP (Dev. Cell, 2017), we have found that Aub directly binds *Cbl* mRNA; it is therefore unlikely that the lack of CCR4 would affect Aub binding to *Cbl* mRNA. GFP-Aub iCLIP in Ma et al. were performed in cultured GSCs and show direct binding of Aub to *Cbl* mRNA in GSCs (Figure 5). Using these GFP-Aub iCLIP, we have found that *nanos*, *mei-P26* and *fused* mRNAs are also directly bound by Aub in GSCs. This information is shown in Appendix Figure S2.

5) In Fig 4G, for clear conclusion for Cbl mRNA deadenylation-independent regulation, it is required to include a reference mRNA which is degraded by deadenylation (and the cancellation by twin mutant). *bam*[delta86] had much less signal. Does this mean Cbl mRNA is low in GSC tumor? The control gene is needed for this analysis. In addition, to show the deadenylation event strictly in GSC, *aub* or twin mutation should be combined with *bam* mutant (GSC tumor). How can the authors identify A12 position? The size ladder is needed.

We have now included a positive control in ePAT assays, *mei-P26* mRNA, which we have shown previously to be deadenylated by CCR4 (Joly et al., Stem Cell Reports, 2013). These data are shown in Figure EV4B. Whether CCR4-NOT can induce deadenylation or translational repression independent of deadenylation might depend on other RNA binding proteins in the complex. This point is discussed in the Discussion p. 16.

We have included more ePAT assays in *bam* mutant GSCs in Source Data For Figure 4G. These data do not indicate a lower expression of *CblL* mRNA in *bam* mutant.

We have tried twice to generate *aub*, *bam*^{Δ86} double mutants, using two different genetic protocols, but these flies were not viable. Moreover, we have shown previously that *twin bam* double mutant GSCs differentiate, and thus *twin bam* double mutant ovaries do not contain GSCs only (Joly et al., Stem Cell Reports, 2013).

The A12 position corresponds to the basal level of the smear according to the ePAT assay protocol (the anchor primer contains 12 T). We have now included the size ladder (Figure 4G).

6) Fig S5, 5A-D; the authors tested immunostaining of Cbl in twin mutant. The effect was much weaker than Aub mutant clone. The reduced Cbl level is not evident in WT GSC compared to CB. This is inconsistent with Fig. 5D. Additionally, I do not see the difference of 8C4 signal between GSC and CB in Fig 5A. Fig 5D needs more sample number (it is currently only 8) like other analyses.

We have now performed additional Cbl staining in *twin* mutant germaria; they show increased levels of Cbl protein in *twin* mutant and are presented in Figure EV4F-G.

We have performed more Cbl staining in wild-type germaria, which confirmed a transient increase in Cbl levels in cystoblast as compared to GSCs. We have now quantified the percentage of cystoblasts showing this increase (Figure 6D). We have increased the numbers for quantification

of *Cbl* levels in cystoblasts and GSCs (Figure 6E).

7) Fig 6C and Fig 3G; can *cbl*[EY] or *cbl*[MB] mutation rescue the GSC loss phenotype in aged twin mutant? (*twin*[DG24102]/*cbl*[EY or MB]/ *twin*[DG24102])

This experiment requires the generation and analysis of *twin Cbl* double mutant that would take 5 to 6 months. This is beyond the time frame that was allocated to us for the re-submission of the manuscript. In addition, *twin* is a general factor of mRNA regulation: it represses mRNAs in GSCs through interaction with Pumilio and Nanos. It is also very likely to repress another set of mRNAs through interaction with the microRNA machinery. Therefore, even though *Cbl* mutant rescues the GSC loss phenotype of *aub* mutant, its ability to rescue the GSC loss phenotype of *twin* mutant is not expected.

Minor points

1) In Fig1E, does the green color show GFP-Aub or Vasa (the legend says Vasa for green)?

This is GFP. Thank you for pointing out this mistake, we have corrected it.

2) Description format of the transheterozygous genotype is not unified, like *aub*[HN2/QC42] versus *aub*[HN2]/*aub*[QC42].

We have corrected this.

Referee #3:

The manuscript by Rojas-Rios et al. makes a substantial contribution to our understanding of *Drosophila* germline stem cell (GSC) biology. It identifies *Cbl* as a novel factor that promotes GSC differentiation. Furthermore, it defines a role for Aubergine (Aub) in GSC maintenance, which it carries out by repressing *Cbl* translation by recruiting the CCR4-NOT to the *Cbl* 3' UTR.

In general I find the experiments are carefully performed and convincingly support the conclusions the authors wish to draw. I do however have a few specific concerns.

1. Most importantly, while substantial evidence is presented that piRNAs in general are required for Aub-mediated regulation of *Cbl*, no hypothesis is presented as to what the specific role of these piRNAs might be. For example, does *Cbl* mRNA have sites of complementarity with some Aub-associated piRNA, that might contribute to recruitment of Aub? The significance of the work would be greater if this question were better resolved.

As also indicated in point 2 from Referee#2, we have used the data set of GFP-Aub iCLIP in cultured GSCs (Ma et al., *Dev. Cell*, 2017) to identify the sites of Aub interaction in *Cbl* mRNA. We then identified piRNAs complementary to *Cbl* mRNA in ovarian and embryonic piRNA libraries. Strikingly, we found abundant piRNAs in close proximity to Aub crosslink sites. These data are shown in Figure 5.

2. In Fig 3C the result for NOT1 is very clear but the co-IP of CCR4 with GFP-Aub is not much above the (high) background signal. Can a different CCR4 antibody be tried for this experiment?

We agree with this comment, however, to our knowledge, no other *Drosophila* anti-CCR4 antibody is available. We show in Figure 3C, NOT1 co-IP with Aub. NOT1 is the scaffold subunit in the complex, which in many cases interacts with RNA binding proteins to recruit the whole complex to target mRNAs. Therefore, co-IP of NOT1 with Aub is a relevant result. We have now tested an additional subunit, NOT3. This new result is shown in Figure 3D.

3. In Fig 4G I don't understand what is going on with *Cbl* poly(A) tail length in the *bam* mutant as there is very little signal at all. Nor do I understand how this result supports the sentence on page 12 that begins "ePAT assays from *bam*[Δ 86] ovaries..."

We have now provided a new data for *Cbl* mRNA ePAT assay in Figure 4G, in which ePAT in *bam* mutant is more visible. Other examples of *Cbl* ePAT in *bam* mutant are shown in Source Data For Figure 4G.

4. Also with regard to Fig 4G to my eye the average poly(A) tail length looks longer in the *aub* lane than in wild-type and in *twin*. As this is contrary to the conclusion as stated in the text, this

experiment should be repeated multiple times and the results quantified.

We have now provided several examples of *Cbl* ePAT assays in *aub* and *twin* mutants, in Figure 4G and Source Data For Figure 4G. The small variations observed are in the range of variations for this experiment and are also visible between different examples of the same genotype. We conclude that there is no defect in *Cbl* poly(A) tail length in *aub* and *twin* mutants. In contrast, the effect of *twin* in lengthening the poly(A) tail of the control *mei-P26* mRNA is visible (Figure EV4B).

I hope that you will find this revised version satisfactory.

I thank you very much in advance and I am looking forward to hearing from you.

Sincerely yours.

Martine Simonelig

Thank you for submitting a revised version of your manuscript. It has now been seen by two of the original referees whose comments are shown below.

As you will see the referees both appreciate the work you have done to address the original criticisms, although they also mention a few points that will need additional discussion/clarification before the final manuscript can be accepted for publication. You will see that ref #1 would ideally have liked to see new experimental data on the roles for dFMR1 and Twin in the pathway described here and that this person also still has some concerns about the Chk2-rescue experiment. I realise that addressing the broader functional context for Aub in GSC growth and cell renewal is beyond the scope of the current revision and the dFMR1/Twin data will therefore not be a requirement from our side. However, I would encourage you to provide additional discussion of the points raised by ref #1. For ref #2, this person is generally satisfied with the revision and only asks for quantification of the GFP-Aub enrichment on the Cbl mRNA.

Based on this input from the referees I would like to invite you to submit a final version of the manuscript in which you address the remaining concerns as listed above.

Thank you again for giving us the chance to consider your manuscript for The EMBO Journal, I look forward to receiving your final revision.

REFeree REPORTS

Referee #1:

In this revised version of their manuscript, the authors have addressed several of the main points raised by the referees. Overall, I find that the strong points and the weak points of the manuscript remain the same.

My major criticisms were the following:

- 1) Aubergine in GSCs and chk2 rescue: (Ma et al., Dev Cell 2017) published that removing chk2 could rescue GSCs loss and germ cell differentiation defects induced by aubergine mutations. In contrast, Simonelig and colleagues had found in the previous version, that chk2 mutation could only rescue differentiation defects but not GSC loss. In this version, by quantifying differently the number of GSCs, they found that chk2 could partially rescue the loss of GSCs (in addition to differentiation defects). It is reassuring that these results are closer to published results. However, it is hard to reach any strong conclusion with a "partial" rescue...To me, this conclusion remains unresolved.
- 2) Cbl and GSCs. Cbl had been involved in GSC maintenance, together with FRM1 ; and this work was not mentioned in the previous version. The authors now cite this work, but do not address experimentally how these previous results fit with their own results.
- 3) Twin and GSCs. Similarly, to point 2, several interactions were previously reported for Twin/Aubergine and transposons (Fu et al. 2015), and not mentioned in the previous version. These results are now discussed in the novel version of the MS, but no experimental work has been done to integrate these results with the authors' model.
- 4) Fluorescent quantification of cbl expression in GSC vs Cystoblasts. As in the previous version: all differential expression of proteins (Cbl, Aubergine, AubAA, etc...) are quantified by measuring fluorescent intensity, which can be very variable. It remains the least convincing data of the manuscript.

My specific comments were:

- 1) Aub localization in ago3 mutant germ cells: the authors answered convincingly on Figure EV2.
- 2) ePAT assay was not clear: it is much improved now.

Referee #2:

In general, the authors nicely answered to my points with additional data that support their conclusions coherently. As the critical response to my biggest concern about the piRNAs involved in, the new Fig 5 identified a group of piRNAs complementary to *Cbl* mRNA sequences. These results are quite complementary to that Aub-AA devoid of piRNA loading cannot complement the defects of GSC in Aub mutants. Given the fact that their target regions are close proximity of the Aub-binding region, it is quite reasonable to think that these piRNAs are functionally involved in Aub-mediated *Cbl* mRNA regulation. However, Aub seems to bind 3'UTR of *CblS* (binding sites are much more than those in *CblL*), where no obvious piRNAs were mapped. In addition, since the authors did not show normalized counts or peaks of GFP-Aub iCLIP, it is impossible to see how much GFP-Aub is bound to *Cbl* RNA. The authors should provide such data (for *CblS* and *L*), and any statement or argument for the binding sites and/or enrichment in *CblS* RNA in Results or Discussion.

The authors also added a new data that Aub-AA retains the ability to bind CCR4-NOT complex (new Fig 3D), which pinpoints the prerequisite of piRNAs loaded onto Aub for *Cbl* regulation. I think all of these data collectively constitute the concrete evidence supporting the model proposed in Fig 7. The new Fig 6D also gives an important information that *Cbl* downregulation is transient and not constantly down-regulated in GSC population. It can explain why in some panels we do not see the difference of 8C4 signals between GSC and cyst cells. It may reflect the delicate *Cbl* regulation by Aub-piRNA in order to promise the proper GSC self-renewal. Related to the ePAT assay, the authors introduced *mei-P26* mRNA as positive control (supp data related to new Fig 4G) and the result is much more convincing than before. One of the remaining questions that should be addressed in future would be how Aub-piRNA complex can induce such GSC-specific repression on *Cbl* mRNA.

Overall, I think the current version of this manuscript could be published in EMBO Journal after the abovementioned issue is properly addressed.

2nd Revision - authors' response

30 August 2017

I thank you very much for your decision regarding our manuscript number EMBOJ-2017-97259R, "Aubergine and piRNAs repress the proto-oncogene *Cbl* for germline stem cell self-renewal" by Rojas-Rios et al. We are submitting a revised version of the manuscript in which we have addressed the remaining concerns of the referees, as well as the editorial issues as follows.

As requested by Reviewer #1, we have clarified the Discussion about i) the partial rescue of *aub* mutant GSC loss by *Chk2* mutant (p. 17, first paragraph), ii) the potential role of CCR4 in transposable element regulation (p. 17, second paragraph), and iii) the possible interaction between Aub and dFMR1 for *Cbl* mRNA regulation (p. 19).

Reviewer #2 requested that we showed the quantification of Aub crosslinks on *Cbl* mRNAs, and the mapping of complementary piRNAs to the short form of *Cbl* mRNA (*CblS*). These data are now shown in Figure 5A (top panel) and Figure 5E. We have mentioned the lower targeting by piRNAs in *CblS* 3'UTR in the Results Section (p. 13).

I hope that you will find this revised version satisfactory.

I thank you very much in advance and I am looking forward to hearing from you.

3rd Editorial Decision

04 September 2017

Thank you for submitting the final revision of your manuscript, I'm please to inform you that the study has now been accepted for publication in The EMBO Journal.

Corresponding Author Name: Martine Simonelig
 Journal Submitted to: EMBO Journal
 Manuscript Number: EMBOJ-2017-97259R